

# CTDAS-Lagrange v1.0: A high-resolution data assimilation system for regional carbon dioxide observations

Wei He[1,2], Ivar R. van der Velde[3,4], Arlyn E. Andrews[3], Colm Sweeney[3,4], John Miller[3], Pieter Tans[3], Ingrid T. van der Laan-Luijkx[5,6], Thomas Nehrkorn[7], Marikate Mountain[7], Weimin Ju[1], Wouter Peters[2,5], Huilin Chen[2,4]

[1]International Institute for Earth System Science, Nanjing University, Nanjing, China
[2]Center for Isotope Research (CIO), Energy and Sustainability Research Institute Groningen (ESRIG), University of Groningen, Groningen, 9747 AG, The Netherlands
[3]Global Monitoring Division, NOAA Earth System Research Laboratory, Boulder, Colorado, USA
[4]Cooperative Institute for Research in Environmental Sciences (CIRES), University of Colorado, Boulder, Colorado, USA
[5]Department of Meteorology and Air Quality, Wageningen University, Wageningen, The Netherlands
[6]Utrecht University, Institute for Marine and Atmospheric Research, Utrecht University, the Netherlands
[7]Atmospheric and Environmental Research, Lexington, MA, USA

Correspondence to: Huilin Chen (Huilin.Chen@rug.nl)

**Abstract.**

We have implemented a regional carbon dioxide data assimilation system based on the CarbonTracker Data Assimilation Shell (CTDAS) and a high-resolution Lagrangian transport model, the Stochastic Time-Inverted Lagrangian Transport model driven by the Weather Forecast and Research meteorological fields (WRF-STILT). With this system, named as CTDAS-Lagrange, we simultaneously optimize terrestrial biosphere fluxes and four parameters that adjust the lateral boundary conditions (BCs) against $CO_2$ observations from the NOAA ESRL North America tall tower and aircraft Programmable Flask Packages (PFPs) sampling program. Least-squares optimization is performed with a time-stepping ensemble Kalman smoother, over a time window of 10 days and assimilating sequentially a time series of observations. Because the WRF-STILT footprints are pre-computed, it is computationally efficient to run the CTDAS-Lagrange system.

To estimate the uncertainties of the optimized fluxes from the system, we performed sensitivity tests with various a priori biosphere fluxes (SiBCASA, SiB3, CT2013B) and BCs (optimized mole fraction fields from CT2013B and CTE2014, and an empirical data set derived from aircraft observations), as well as with a variety of choices on the ways that fluxes are adjusted (additive or multiplicative), covariance length scales, biosphere flux covariances, BC parameter uncertainties, and model-data mismatches. In pseudo-data experiments, we show that in our implementation the additive flux adjustment method is more flexible in optimizing NEE than the multiplicative flux adjustment method, and that the CTDAS-Lagrange system has the ability to correct for the potential biases in the lateral boundary conditions and to resolve large biases in the prior biosphere fluxes.





Using real observations, we have derived a range of estimates for the optimized carbon fluxes from a series of sensitivity tests, which places the North American carbon sink for the year 2010 in a range from -0.92 to -1.26 PgC/yr. This is comparable to the TM5-based estimates of CarbonTracker (version CT2016, -0.91 ± 1.10 PgC/yr) and CarbonTracker Europe (version CTE2016, -0.91 ± 0.31 PgC/yr). We conclude that CTDAS-Lagrange can offer a versatile and
computationally attractive alternative to these global systems for regional estimates of carbon fluxes, which can take advantage of high-resolution Lagrangian footprints that are increasingly easy to obtain.

## 1 Introduction

$CO_2$ exchange between the terrestrial biosphere and the atmosphere has a strong impact on the climate system, which makes it crucial to quantify the amount of $CO_2$ exchange, and to better understand the interactions between the global carbon cycle
and climate change. Atmospheric measurements of trace gas mole fractions provide constraints for the estimates of biosphere surface fluxes from regional to global scales, and complement bottom-up biosphere modeling that typically targets site to ecosystem scales in the earth system. Inferring biospheric and oceanic surface fluxes from a "top-down" perspective, through an atmospheric inversion, plays an important role in global budgeting efforts (Le Quere et al., 2016), as it takes advantage of the mass-conservation of carbon in the atmosphere and the high-precision measurements done in the atmosphere over the
past decades (Conway et al., 1994; observations are now published through ObsPack available at https://www.esrl.noaa.gov/gmd/ccgg/obspack/index.html).

In the past decade, much attention has been given to estimating carbon fluxes at global scales (e.g. Rödenbeck et al., 2003; Peters et al., 2007; Chevallier et al., 2010; Peylin et al., 2013), while regional inversion studies with high spatial resolution
for carbon fluxes are only gaining ground more recently (e.g. Göckede et al., 2010; Schuh et al., 2010; Tolk et al., 2011; Lauvaux et al., 2012; Gourdji et al., 2012; Broquet et al., 2013; Shiha et al., 2014; Alden et al., 2016; Kountouris et al., 2016; Feng et al., 2016). Such regional inversion studies contribute to a better understanding of the mechanism through which carbon fluxes react to environmental variations at a fine scale. But to link carbon fluxes and environmental drivers to atmospheric measurements, a high-resolution transport model is typically needed. High-resolution Lagrangian models are
more computationally efficient than traditional Eulerian models, which makes them suitable for computation-intensive regional inversion studies.

However, both global and regional inversion studies suffer from various uncertainties, including transport and representation errors, possible observational biases when data from different laboratories are combined, and uncertainties in a priori fluxes.
For regional inversions, errors in lateral boundary conditions (BCs) become another critical issue (Alden et al., 2016; Gerbig, et al., 2003; Schuh et al., 2010; Lauvaux et al., 2013), and can bias flux estimates, particularly for smaller areas (Peylin et al., 2005; Göckede et al., 2010) and for shorter period (Peylin et al., 2005). Several methods to create lateral boundary





conditions have been employed, including deriving them from mole fraction fields of global inversions (Kountouris et al., 2016) and in situ mole fraction observations, e.g. aircraft profiles or satellite observations (Jiang et al. 2015). Embedding a regional inversion inside a global model domain has been widely applied for $CO_2$, $CH_4$ and $N_2O$ flux estimates (Bergamaschi et al., 2010; Corazza et al., 2011) for example within the nested TM5 model framework. Gourdji et al. (2012) compared an

empirical BC derived from aircraft profiles and marine boundary layer data with BC values taken from CarbonTracker CT2009 optimized mole fraction fields, and pointed out the former might be more accurate than the latter. Various studies apply aircraft measurements to correct model-derived BCs either before (Broquet et al., 2013), or during regional inversions (Lauvaux et al., 2012; Brioude et al., 2012; Wecht et al., 2014). Adjusting BCs using the inverse modeling framework is desirable as it guarantees consistency between all sources of information used. Recently, Jiang et al. (2015) assimilated the

MOPITT satellite profile data to optimize BCs during the estimation of North American CO emissions, and reported a reduction of the mean residual bias in the posteriori simulation (simulations minus observations) from -13.3% to 3.5%.

To better understand regional carbon fluxes, we developed a data assimilation system that employs a high-resolution Lagrangian atmospheric transport model, the WRF-STILT model. Our assimilation system, the CarbonTracker Data

Assimilation Shell – Lagrange (referred to as "CTDAS-Lagrange"), is based on the CarbonTracker Europe system, which is a widely-applied global inversion system (Peters et al., 2010; van der Laan-Luijkx et al., 2015; van der Laan-Luijkx et al., 2017). In our new system, we simultaneously optimize biospheric fluxes and the lateral BCs using tower and aircraft observations. We use a priori biosphere fluxes from the SiBCASA biosphere model (Schaefer et al., 2008), and the other a priori fluxes for the components ocean, fossil fuels, and fires are from CT2013B (accessible from the archived release

https://www.esrl.noaa.gov/gmd/ccgg/carbontracker/CT2013B/). $CO_2$ observations come from NOAA Programmable Flask Package (PFP) data from tall towers and aircraft sites. Aircraft observations above 3000 m and tower observations are used to optimize BCs while only tower observations were used to optimize the terrestrial biosphere fluxes at the surface. We investigate the impact of different a priori fluxes and BCs, two alternative ways of adjusting fluxes (additive and multiplicative), covariance length scales, BC parameter uncertainties, model-data mismatch, and observational data choice

on the optimized fluxes.  Based on the above investigations, we have constructed a range of estimates, and then compared the inversion results with those of contemporary inversion studies.

The purpose of this paper is to describe and demonstrate the CTDAS-Lagrange data assimilation system. We have performed preliminary inversions using a subset of the available $CO_2$ data for North America for a single year.  A more comprehensive

analysis is planned that will incorporate additional datasets and cover a longer time period. This paper is organized as follows: in Sect. 2 we introduce the modeling framework and observation data used for this study, Sect. 3 presents results of the system performance and sensitivity runs, followed by discussion and conclusions in Sect.4.



## 2 Data and model

### 2.1 CO$_2$ observations

Our system assimilates atmospheric CO$_2$ mole fraction measurements made from the NOAA ESRL Global Greenhouse Gas
Reference Network, specifically, the analysis results of air samples collected by automated flask-sampling systems that are
known as programmable flask packages (PFPs). The advantage of using PFP flask data is that more than 50 compounds,
including carbon monoxide, are also available together with CO$_2$ measurements. The data collected in North America at 8
tall tower sites and 6 selected aircraft sites in 2010 are used for this study. The air samples were collected daily or on
alternate days during mid-afternoon at the tall tower sites (Andrews et al., 2014), and biweekly or monthly at the selected
aircraft sites (Sweeney et al., 2015). The location of the observations is shown in Fig. 1. The data are provided to the model
input as an ObsPack (Masarie et al., 2014).

#### 2.1.1 Tall tower observations

Detailed site and sampling information of the tall tower observations is listed in Table 1. Andrews et al. (2014) used flask
versus in situ comparisons for quality control and pointed out such comparisons suffer from quasi-continuous in situ data
(due to e.g. switching of sampling lines among different heights, calibrations), difference in sampling time, and atmospheric
variability. The mean differences between PFP flask and in situ CO$_2$ measurements over all 8 sites for 2010 range from 0.08
to 0.32 ppm, with the standard deviation of the differences for each site ranging from 0.2 to 0.6 ppm and increasingly
positive differences over the period 2008-2011. According to Andrews et al. (2014), the mean differences are likely caused
by potential biases in an increasing number of the PFP flask measurements that may result from contamination caused by
routine use throughout the network or by use under polluted conditions. The flask sampling protocol has since been changed
such that the flask is pressurized with ambient air prior to sample collection and held at high pressure for several minutes
then vented and resampled. Agreement has improved between flask and in situ measurement systems so the difference is
reliably better than 0.2 ppm.

#### 2.1.2 Aircraft profiles

The NOAA ESRL aircraft CO$_2$ profile data (Table 1) are used to optimize lateral boundary conditions. The Global
Greenhouse Gas Reference Network's aircraft program (Sweeney et al., 2015) has been collecting air samples for vertical
profile measurements over North America since 1992. For each individual flight, 12 flask samples are collected from 500
meters above ground up to 8000 meters above sea level at most aircraft sites. Of 15 ongoing aircraft sites by 2014, we have
selected 12 sites (8 close to the domain boundary, and 4 in the middle of the domain) for this study. Because the aircraft
program uses the same PFP flasks as in the tall tower program, the aircraft CO$_2$ measurements may have potential biases as
well. Indeed, Karion et al. (2013) report PFP flask minus in situ CO$_2$ measurements of $0.20 \pm 0.37$ ppm for the aircraft





measurements over Alaska from 2009 to 2011, a similar magnitude of biases as found in the tall tower PFP flask versus in situ comparisons.

### 2.1.3 Data filtering

We use daytime data from the tall towers that are collected between 10:00 and 18:00 local time to constrain surface fluxes.
Aircraft observations made at altitudes higher than 3000 meters above ground at all hours are used to constrain boundary conditions. In CTDAS-Lagrange, we use fossil fuel emissions based on inventory estimates and do not attempt to optimize them. We remove $CO_2$ observations that are likely strongly influenced by fossil fuels before optimizing biosphere fluxes. This diminishes the potential biases in optimized biosphere fluxes that are caused by local fossil fuel sources and/or by representation errors in the simulated fossil fuel $CO_2$ signals. To achieve this, we use CO measurements as a proxy for fossil
fuel influences, realizing that especially in summer other sources of CO can contribute to enhanced mole fractions. We first calculate CO enhancement as the difference between the CO observation and the background value, i.e. the corresponding value from a second order harmonic function that is fitted to the CO data for each tall tower site. We filter out any $CO_2$ observations with CO enhancements larger than 33.6 ppb, which corresponds to 3-ppm fossil fuel $CO_2$ according to the year-round median apparent ratio of 11.2 ppb/ppm estimated in Miller et al. (2012). About 8.5% of the available $CO_2$ data is
excluded by the CO filter, with the majority coming from the two sites STR and WGC in California.

### 2.2 The CTDAS-Lagrange system

The CTDAS-Lagrange system aims to improve the estimates of regional carbon fluxes by combining a high spatial resolution Lagrangian modeling framework with the existing Data Assimilation Shell (van der Laan-Luijkx et al., 2017). Transport of atmospheric $CO_2$ in the main application of CTDAS: the CarbonTracker Europe system, is realized by using the
global, two-way nested transport model TM5 (3°x2° global, and 1°x1° for one or more regional domains of interest), driven by 3h meteorological parameters. The CTDAS-Lagrange system replaces the coarse TM5 transport model with a Lagrangian transport model with high spatial resolution. Another advantage of the CTDAS-Lagrange system is its significantly improved time efficiency. Outputs from the Lagrangian transport model can be stored as measurement footprints (influence functions) so that the $CO_2$ mole fractions resulting from different surface flux configurations can be simulated offline
afterwards, using simple matrix multiplications rather than full transport calculations. In addition, these stored outputs can be used for other species directly, reducing significantly the computation time when performing multiple/other species inversions, i.e. for the extension of our system to multi-species applications.

### 2.2.1 Atmospheric transport model

The Stochastic Time-Inverted Lagrangian Transport model coupled with the Weather Forecast and Research (WRF-STILT)
is employed in our system (Lin et al., 2003; Nehrkorn et al., 2010). The STILT model is a receptor-oriented framework that links surface fluxes of trace gases with atmospheric mole fractions. During a WRF-STILT run, an ensemble of particles is





released at the observation location (receptor) at a certain time, and particles are transported backward driven by the WRF wind fields. The influence function, i.e. footprint, for that particular receptor and time can be computed based on the density of the particles in the surface layer defined in STILT as the lower half of the well-mixed boundary layer.

We leverage a footprint library created for the NOAA CarbonTracker Lagrange regional inversion framework (https://www.esrl.noaa.gov/gmd/ccgg/carbontracker-lagrange/). The WRF-STILT model was run with 500 particles that are traced backward for 10 days. The WRF model was run for North America, with the spatial resolution of 10 km for the inner domain (~ 25 – 55 $^{o}$N; 135 – 65 $^{o}$W) and 30 km for the outer domain (~ 10 – 80 $^{o}$N; 170 – 50 $^{o}$W). STILT footprints are stored with $1^{o}$ x $1^{o}$ spatial resolution and hourly time resolution. Snapshots of the 3-dimensional particle distribution are also

stored to enable assignment of boundary values according to where particles intersect with the domain of the inversion.

**2.2.2 Optimization scheme**

In the CTDAS-Lagrange system, we extended the existing Ensemble Kalman Smoother method as is implemented in CarbonTracker and CarbonTracker Europe (Peters et al., 2005, 2007, 2010; van der Laan-Luijkx et al., 2017) to simultaneously optimize biosphere fluxes and boundary condition parameters.

We use two alternative ways of adjusting the total surface fluxes (additive and multiplicative), while simultaneously optimizing the lateral boundary conditions by optimizing four parameters that are implemented as follows:

$$C(X_r,t_r) = C_0(X_r,t_r) + \sum_{i=1}^{4} W_i * \beta_i + S(x,y,t) * \begin{Bmatrix} f[\lambda, F_{bio}(x,y,t)] \\ F_{ff}(x,y,t) \\ F_{ocn}(x,y,t) \\ F_{fire}(x,y,t) \end{Bmatrix} \quad \text{(1)}$$

where $C(X_r,t_r)$ is the simulated $CO_2$ mole fraction [ppm] at the location of the observation (receptor) $X_r$ and time $t_r$, $C_0(X_r,t_r)$

refers to the contribution of advection from the lateral boundary condition [ppm]; $\beta_i$ [ppm] is adjusted to optimize the lateral boundary condition for each of the four sides of the regional domain for each 10-day period. The boundary condition mole fraction $\beta_i$ is weighted by the pre-calculated coefficient $W_i$ [unitless] that is determined as the ratio of the number of particles exited from one side of the domain to the number of particles exited from all sides of the domain. The domain considers 3000 m as the top boundary, i.e. the particles that exited the domain below 3000 m are not considered. This choice reflects

the dominant influence of surface fluxes over lateral advection for particles that spent considerable time within the inner domains. $S(x,y,t)$ is the footprint (sensitivity of mole fraction variations to surface fluxes, [ppm/(μmol m$^{-2}$ s$^{-1}$)]) calculated with STILT. Biosphere fluxes $F_{bio}(x,y,t)$ [μmol m$^{-2}$ s$^{-1}$] are optimized by either additively or multiplicatively optimizing a set





of parameters $\lambda$ [$\mu$mol m$^{-2}$ s$^{-1}$, or unitless] for each 1x1 degree grid in the domain for each 10-day period, represented by the function $f[\lambda, F_{bio}(x,y,t)]$ in the equation. $F_{ocn}(x,y,t)$, $F_{ff}(x,y,t)$, and $F_{fire}(x,y,t)$ denote the CO$_2$ fluxes [$\mu$mol m$^{-2}$ s$^{-1}$] exchanged with ocean, from fossil fuels and fires, and these are fixed.

The state variables therefore include the gridded adjusting parameters for the biosphere fluxes (3078 land grids with 1x1 degree resolution over North America) plus those for the four boundary condition parameters, leading to a total of 3082 parameters for each 10-day period. The state variables are optimized simultaneously within each period. Considering that aircraft observations above 3000 meters contain mostly information about boundary conditions and have low or even no sensitivity to surface fluxes, we optimize the biosphere fluxes using tower observations only, and optimize boundary

conditions with both tower and aircraft observations. This separation is applied through localization of the Kalman Gain matrix.

### 2.2.3 System setup

The system aims to optimize (non-fire) net ecosystem exchange (NEE) of CO$_2$ between biosphere and atmosphere, and requires prior biosphere fluxes, lateral boundary conditions, and other fixed fluxes such as fossil fuel emissions, ocean and

fire fluxes as model input. This section describes the setup of the base case run.

We use biosphere fluxes simulated by the combined Simple Biosphere and Carnegie-Ames-Stanford Approach (SiBCASA) model (Schaefer et al., 2008) as a prior and fixed fossil fuel burning, ocean, and fire fluxes from CT2013B (Peters et al. 2007, with updates documented at http://carbontracker.noaa.gov) with the latter two being negligibly small in the annual

mean over our domain of interest, but still included since they introduce spatiotemporal variations in CO$_2$ mole fractions. More details about these prior and the component fluxes are described in section 2.2.4. The lateral boundary condition is also taken from the 4-D mole fraction fields simulated by CT2013B. We use the WRF-STILT transport model. Here we give a short summary of the configuration of the base case run: we prescribe an additive biosphere fluxes uncertainty of 1.6 $\mu$mol m$^{-2}$ s$^{-1}$; a prior uncertainty of the boundary condition parameters of 2.0 ppm, a model-data-mismatch of 3 ppm for surface

sites and 1 ppm for aircraft sites, and a covariance length scale of 750 km.

We estimate the additive flux adjustments for each grid box in our domain, but a covariance structure is used to reduce the number of degrees of freedom in the state vector, and to balance it with the number of available observations. The covariance is calculated as an exponential function that decreases with distance between grid boxes, using a decorrelation

length scale of 750 km. This covariance is only used between grid boxes that have the same dominant plant-functional type, as specified though the ecoregion maps that are also used in CT2013B. These in turn are based on TransCom regions, as well as the Olson ecosystem classification (Olson et al., 2002). Where CT2013B uses single scaling factors for each ecoregion,




our gridded approach has approximately 122 degrees of freedom within its 3078 additive adjustment parameters as compared to an average of 112 independent observations per assimilation time step.

We have adapted the fixed lag Ensemble Kalman Smoother method from Peters et al. (2005) to estimate fluxes and BC per 10-day time step. Because the footprint of each receptor can go back in time up to 10 days, we need a total assimilation window of 20 days to account for the backward trajectories that overlap two time steps. Therefore, the total state vector contains flux and BC parameters for two 10-day time steps (3082 x 2). The time stepping cycle works as follows (see Fig. 2): First, we use an ensemble of parameters derived from the total state vector to calculate an ensemble of modeled $CO_2$ mole fractions for each measurement extracted in the current 10-day time step. These state vector parameters reflect the influence of fluxes and boundary conditions on the modeled $CO_2$ in the current 10-day time step and the previous 10-day time step that has already been optimized once in the previous cycle. In the next step, the set of Ensemble Kalman Smoother equations as outlined in Peters et al. (2005) is solved to give a new set of optimized state vector parameters and its ensemble, where the state vector of the previous time step is optimized for a second and final time. Modeled $CO_2$ from the previous time step is updated using the final state vector. Finally, the next cycle starts 10 days forward in time by introducing a new set of measurements. In this way, each 10-day state vector is finalized after two cycles of optimization.

A comparison of the setup between the base case and sensitivity runs (described in the section 2.3) is given in Table 2.

### 2.2.4 Prior biosphere and other fluxes

The prior biosphere fluxes are simulated by the SiBCASA model, a diagnostic biosphere model, which combines photosynthesis and biophysical processes from the Simple Biosphere (SiB) model version 3 with carbon biogeochemical processes from the Carnegie-Ames-Stanford Approach model (Schaefer et al., 2008). Meteorological driver data is provided by the European Centre for Medium-Range Weather Forecasting (ECMWF). SiBCASA calculates the surface energy, water, and $CO_2$ fluxes at a 10-minute time step on a spatial resolution of 1° x 1°, and predicts the moisture content and temperature of the canopy and soil (Sellers et al., 1996). We use the 3-hourly mean $CO_2$ fluxes as is used in van der Velde et al. (2014) and van der Laan-Luijkx et al. (2017).

CT2013B presents a multi-model prior suite of inversion, which includes two different flux data sets for ocean and fossil fuels, respectively. The fire fluxes are based on the Global Fire Emissions Database (GFED) 3.1, which are calculated with the CASA model (Giglio et al., 2006; van der Werf et al., 2006). The fire fluxes are not optimized in CT2013B. The two different prior ocean fluxes for CT2013B include a long-term mean of ocean fluxes that is derived from the ocean interior inversions (Jacobson et al., 2007) and a climatology data set that is created from direct observations of seawater around the world and was interpolated onto a regular grid map using a modeled surface current field (Takahashi et al., 2009). We use the optimized ocean fluxes of CT2013B that are calculated as the mean of an ensemble of run results. The two different





fossil fuel fluxes for CT2013B are the "Miller" emissions dataset and the "ODIAC" emissions datasets (Oda and Maksyutov, 2011). The difference of the two data sets is the processing schemes on the totals, spatial distribution, and temporal distribution of fossil fuel emissions. The fossil fuel fluxes are not optimized in CT2013B. We use the fixed fossil fuel data of CT2013B, which is an average of "Miller" and "ODIAC". The final product of these fluxes is provided on 1° x 1° degree

grid at a 3-hourly temporal resolution. More details can be found at the CarbonTracker website CT2013B (https://www.esrl.noaa.gov/gmd/ccgg/carbontracker/CT2013B/index.php).

### 2.3 Sensitivity runs

#### 2.3.1 Lateral boundary conditions

The lateral boundary conditions could be constructed either from interpolated measurements or from the output of a global

tracer model. The base case uses CT2013B optimized mole fraction fields.

To study the impact of different lateral boundary conditions on flux optimization, we have tested optimized mole fraction fields with the spatial resolution of 1° x 1° from CarbonTracker Europe (CTE2014), as well as empirical background mole fraction fields (EMP). The empirical background mole fraction fields are derived from marine boundary layer observations

and aircraft vertical profiles along the west coast of North America, and are constructed and used in a similar way as in Miller et al. (2013) and Hu et al. (2015). They are interpolated daily $CO_2$ mole fractions, with spatial resolution of one degree of latitude at eight levels of height (500m, 1500 m, 2500 m, 3500 m, 4500 m, 5500 m, 6500 m, 7500 m).

We also assign different prior uncertainties other than 2 ppm for the boundary condition parameters. Experiments are

designed as follows:

BC1: using CTE2014 mole fraction fields as lateral boundary conditions;

BC2: using EMP as lateral boundary conditions;

Pbc1: set the uncertainty of the boundary condition parameter to 1 ppm;

Pbc2: set the uncertainty of the boundary condition parameter to 3 ppm.

For all other aspects, the model is configured to be identical to the base case run.

#### 2.3.2 Prior biosphere fluxes

Gurney et al. (2004) point out that inversion results can be sensitive to a priori fluxes for regions with sparse observations

while the fluxes can be well constrained by areas with dense observations. To investigate the impact of different a priori fluxes on the optimized fluxes, we have designed two sensitivity runs that incorporate two alternative biosphere fluxes as a priori fluxes as follows:





B1: SiB3 biosphere fluxes,

B2: CT2013B optimized biosphere fluxes.

Except for the difference in the a priori biosphere fluxes, the two sensitivity runs share the same model setup as the base
case. SiB3 biosphere fluxes are the simulation results of the third version of the Simple Biosphere model (Baker et al.,
2008), with hourly fluxes and a spatial resolution of 1° x 1°. CT2013B optimized biosphere fluxes are the outputs of
CT2013B that optimizes the global surface biosphere fluxes, which uses higher-resolution transport over North America than
other regions. Although these fluxes are already optimized against global and North America $CO_2$ observations, it is still

interesting to optimize them in a different assimilation system, especially when the system employs a high spatial resolution
and different transport model. In addition, CT2013B assimilates a different set of observations compared to CTDAS-
Lagrange. In principle, one expects these fluxes to be most consistent with observations, and to lead to a very similar
posterior mean flux as was prescribed through the prior.

For further analysis of the sensitivity of the CTDAS-Lagrange system to the annual mean and the seasonal magnitude of a
priori fluxes, we have designed a series of runs with modified SiBCASA fluxes. We scaled the respiration of the SiBCASA
fluxes while maintaining the GPP estimate to obtain a priori North American annual mean fluxes ranging from +0.43 to -
2.06 PgC/yr. The tests with prior fluxes +0.43, -0.06, -0.97, -1.44 and -2.06 PgC/yr are labeled with BX1, BX2, BX3, BX4
and BX5, respectively.

**2.3.3 Additive or multiplicative flux adjustment**

The multiplicative flux adjustment of NEE relates the uncertainties to the magnitude of the fluxes. As NEE is the difference
between two gross fluxes gross primary production and ecosystem respiration, 10-day mean NEE can be very small or even
close to zero when GPP and respiration are close to each other, e.g. in the so-called shoulder seasons see Fig. 8, which limits
the ability of using multiplicative flux adjustment to scale the mean fluxes due to low uncertainties in the inversion system

(note that the large diurnal cycle of the net flux will still be scaled though). Scaling both GPP and respiration has been shown
to circumvent this in deriving optimized mean fluxes (Tolk et al., 2011). Here, we have instead implemented both
multiplicative and additive flux adjustment methods. For the multiplicative method, we set the biosphere scaling parameter
variance as 80%, following Peters et al. (2010); for the additive method, the variance is prescribed as 1.6 µmol m$^{-2}$ s$^{-1}$ which
represents the typical flux uncertainty of the multiplicative method during the summer months. Because this value persists

yearlong in the additive run, the total annual uncertainty of this method is higher though. Sensitivity tests for the covariance
are described below. The additive method is used in the base case run, and the multiplicative method was tested as a
sensitivity run.



For a better assessment of the adjusting ability of the two methods, we further perform experimental inversions using pseudo data. We run the CTDAS-Lagrange in a forward mode with the SiBCASA fluxes as prior to generate simulated mole fractions, and then try to recover the "truth" in an inversion using SiB3 fluxes as a priori.

### 2.3.4 Covariance length scales

The covariance length scale determines the rate at which the correlation between the fluxes of two grids within the same ecoregions decreases exponentially with increasing distance. The prescribed covariance effectively reduces the number of unknowns to be solved for, and improves the ability of the inversion system to retrieve optimized fluxes when data are limited (Rödenbeck et al., 2003; Gourdji et al., 2012). To investigate the impact of covariance length scales on optimized fluxes, we performed sensitivity runs with a series of spatial correlation lengths: 300 km, 500 km, 750 km (base case), 1000

km and 1250 km, labeled as CL1 to CL4, respectively.

### 2.3.5 Magnitude of covariance and model-data mismatch

Flux covariance determines the range in which prior biosphere fluxes can be adjusted. It should ideally reflect the uncertainty of prior biosphere fluxes, but information about prior flux errors is not readily available for the priors used here or for terrestrial ecosystem models more generally. To evaluate the possible influence of prior covariances on the optimized

fluxes, we modified the additive uncertainty by ±50%. The model-data mismatch (MDM) is a parameter that describes the capability of our modeling system to match the observations, and is used to deweight observations that are not well represented by the model simulations, e.g. in the case of local influence. The observations are even excluded when the differences between observed and simulated $CO_2$ are larger than 3 times the MDM. We set the MDM to 3.0 ppm for tower sites and 1.0 ppm for aircraft sites. The sensitivity tests that incorporate the covariance and MDM are described below:

Q1: decrease the additive uncertainty magnitude by 50%, which means the covariance is 25% of the default;

Q2: increase the additive uncertainty magnitude by 50%, which means the covariance is 225% of the default;

R1: set the MDM to 2 ppm for tower sites;

R2: set the MDM to 4 ppm for tower sites.

The rest of the model setup is the same as the base case run.

### 2.3.6 Observational data choice

As a sensitivity test, we exclude observations at two tower sites (STR and WGC), which are characterized with larger prior and posterior residuals (simulated minus observed, both mean and standard deviation) than other sites. We have defined one

sensitivity run as follows:





Obs: excluding STR and WGC, and the rest of the model setup is the same as the base case run.

## 3. Results

This section covers the following topics: $CO_2$ mole fraction simulations and seasonal cycles of biosphere fluxes from the
base case run, lateral boundary conditions choice and optimization, sensitivity to a priori fluxes, additive or multiplicative
adjusting parameters, covariance length scales, transport uncertainties, and a summary of ensemble estimates.

### 3.1 Observed and simulated $CO_2$ mole fractions

As an example, a time series of simulated and observed $CO_2$ mole fractions at the LEF tower site for the year 2010 is shown
in Fig. 3. As should be expected from the assimilation, the optimized $CO_2$ records closely follow the $CO_2$ observations over
time, and the optimized residuals (green) are smaller than those from the model forecast (red). The distribution of both prior
and posterior residuals shown on the right side of Fig. 3 indicates improvement from the prior $+0.85 \pm 3.73$ ppm to the
posterior $0.09 \pm 1.55$ ppm. The larger variability of both observed and simulated $CO_2$ between May and November
(compared to the rest of the year) are likely caused by larger variability in the biosphere fluxes during the growing season, as
well as larger variation in atmospheric mixing conditions. A few simulated values (blue) are rejected in the assimilation
procedure because the difference between simulations with prior biosphere fluxes and observations is larger than three times
the assigned model-data-mismatch of 3 ppm for tower sites. For the LEF tower site, 11 out of the total 409, or 2.9% are
rejected, which is slightly larger than the expected rejection rate (based on a 3-σ cut-off for a Gaussian PDF of the errors) of
2% (Peters et al., 2010). It is shown in Table 1 that the rejection rates for most tower sites are around 2-3%, except for WBI
(7.6%) and WGC (18.0%).

### 3.2 Seasonal cycles of net ecosystem exchange of $CO_2$

Fig. 4 shows the seasonal cycle (10-day averages) of net ecosystem exchange of $CO_2$ for the year 2010 for 4 major Olson
aggregated land-cover types of North America (boreal forests/wooded, boreal tundra/taiga, temperate forests/wooded, and
temperate crops/agriculture). The amplitude of the seasonal cycle of temperate forests/wooded is the largest among the four
land-cover types, with both large summertime vegetative uptake and large wintertime respiratory emissions. The
uncertainties of the posterior fluxes have been reduced for all four land-cover types and for almost all seasons of the year,
especially for temperate forests/wooded and temperate crops/agriculture.

The seasonal cycle of the posterior fluxes shows a similar magnitude as the prior. In addition, the optimized fluxes generally
show more fluctuations than prior fluxes over the year, which could be explained by effective constraints from atmospheric
observations and possibly in some cases as artifacts that are caused by the sparseness of the observations. Interestingly, the





temperate crops/agriculture show double troughs in the uptake in May and July-August or a sudden drop in the uptake in June, which could be attributed to early-summer crops/agriculture harvests, temperature anomaly, or drought.

The mean prior and optimized fluxes for the summer months June – August are given in Fig. 5. The optimized fluxes show a
similar spatial pattern as the prior fluxes, but display more spatial details. The optimized results place more carbon uptake in the agricultural Midwest and the forests/wooded in the northeast of the United States, as well as in the boreal forests/wooded and tundra/taiga of Canada; In contrast, less carbon uptake (or carbon emissions) are placed in the Western US, especially in south Utah, north Arizona and Louisiana.

### 3.3 Boundary condition choice and optimization

A comparison of the mole fraction contribution from three lateral boundary conditions for the eight tower sites is summarized in Table 3. The annual means of the CTE2014 are consistently ~0.30 ppm higher than those of the CT2013B for all sites; however, the summertime means of the CTE2014 and the CT2013B are nearly equal except for the two sites AMT and STR. In contrast, the annual means of the EMP and the CT2013B are nearly equal for all sites; however, the summertime drawdowns of the EMP are significantly higher (-1.70 to -0.28 ppm) than those of the CT2013B for all sites
except STR (0.66 ppm). This suggests that the two model-derived BC's provide higher summer background mole fractions than the EMP-based background, which corresponds to a known high bias in summertime $CO_2$ across North America in both versions of CarbonTracker used to construct the BCs.

The optimized annual mean fluxes and the adjustment of the boundary condition parameters for the model runs with
different prior lateral boundary conditions are shown in Table 4. When both biosphere fluxes and BC parameters are optimized, i.e. "Flux+BC" optimization, the optimized annual mean fluxes using three different prior lateral boundary conditions range from -1.26 to -1.08 PgC/yr, with an average of -1.14 ± 0.10 PgC/yr, which have a smaller variation compared to those from the model runs when only biosphere fluxes are optimized, i.e. "Flux only" optimization, that ranges from -1.49 to -1.09 PgC/yr or -1.22 ± 0.23 PgC/yr (discussed in more detail in the following section). The results show that
the additional BC optimization is desired when model-based BCs are used, and that this reduces the annual mean optimized biosphere fluxes by up to 0.23 PgC/yr, or 15.4% of the "Flux only" optimized fluxes. The largest adjustment in the optimized annual mean biosphere uptake takes place in the run with the CTE2014 as lateral boundary conditions, which corresponds to the consistently higher annual means of the boundary condition contributions of the CTE2014 than those of the CT2013B. The contribution of the adjustment of boundary condition parameters to simulated $CO_2$ ranges from -0.18 to
0.16 ppm over all seasons of the year 2010, which stresses that this is a subtle, but systematic, signal to account for in a regional inversion.





The residuals of the model runs with and without BC optimization (not shown), in almost all cases, are significantly reduced after optimization. The reduction in the residuals after optimization for aircraft sites is primarily due to the adjustment of the boundary condition parameters. We notice that the residuals (means and standard deviations) of the model runs with optimized biosphere fluxes and boundary condition parameters for the two sites STR and WGC are larger than those for

other sites. Possible reasons are that the two sites are still significantly influenced by regional fossil fuel signals after the data filtering presented in Section 2.1.3, and are less sensitive to biosphere fluxes (due to their proximity to the West Coast of North America there is less sensitivity to land flux than for other sites). We will investigate the impact of the two observation sites on optimized fluxes in the following section.

The time series of optimized North American averaged biosphere fluxes from the model runs with different prior lateral boundary conditions are shown in Fig. 6. The differences among the optimized fluxes with additional BC optimization (Fig. 6b) become smaller than those from the "Flux only" runs (Fig. 6a). This can also be observed when averaged over major ecoregions (Fig. 6c&d), especially for the boreal forests and temperate forests. The differences in the optimized biosphere fluxes caused by different prior lateral boundary conditions are mostly small, except that the deviation of the EMP optimized

fluxes from the other two is slightly larger for the period July – September.

### 3.4 Sensitivity to prior biosphere fluxes

The optimized annual mean biosphere fluxes and associated BC parameter adjustments from the runs with different prior biosphere fluxes are shown in Table 5. The flux adjustments are in general large, resulting in significantly larger annual mean uptake over North America than the prior; however, the optimized annual mean fluxes from the runs using three

different prior biosphere products are consistent, except for the run using the original CT2013B optimized fluxes. A further check indicates that the residuals of the run are reasonable, but more observations have been rejected compared with the other runs. The rejection takes place in the period from June to August, which is caused by large fluctuations of the a priori fluxes. Note that the a priori CT2013B fluxes are optimized using weekly scaling factors in an assimilation window of 5 weeks long and incur substantial variability (or noise) that averages out over larger scales in CT2013B. But the forward

simulations of the CTDAS-Lagrange system are sensitive to the fluxes and their diurnal cycle only in a 10-day window and therefore more sensitive to this variability (or noise). Therefore, we have made an additional sensitivity test (B2') with smoothed CT2013B fluxes (10-day averaged, identical 3-hourly fluxes across a day in every 10-day period) as a priori, which gives smaller optimized annual fluxes (see Table 5). We hereafter refer to CT2013B-avg for further analysis.

Fig. 7a shows the time series of the North America averaged biosphere fluxes of the model runs with different prior biosphere fluxes. It is noticeable that the difference in the seasonal amplitude between the SiB3 prior biosphere fluxes and the other two prior biosphere fluxes is diminished after optimization. Furthermore, the significant difference among the three prior products for the period August – October is largely reconciled by the inversion. Annual mean fluxes per ecoregion





(Fig. 7c) indicate that the largest adjustment in the fluxes takes place for temperate forests and temperate grass, with fluxes from temperate grass changed from uptake to emissions. Note that the optimized fluxes per ecoregion do not always agree on their magnitudes, which is likely caused by insufficient constraints by observations, especially for the boreal region.

To further investigate the sensitivity of the CTDAS-Lagrange system to the seasonal magnitude and the annual mean of a priori fluxes, we scale the respiration of the SiBCASA fluxes to obtain a variety of a priori fluxes with the annual mean NEE ranging from +0.43 to -2.06 PgC/yr. We find that the seasonal magnitudes of the optimized fluxes are nearly independent of those of the prior fluxes (Fig. 7b), and the range of the annual mean is significantly reduced to -0.9 – -1.45 PgC/yr (Table 6). Like the runs with the prior fluxes from SiBCASA, SiB3 and CT2013B-avg, the optimized fluxes show variations at
multiple times of the year that are a direct result of the corresponding flux adjustment within 10-day windows. The prior/optimized fluxes per ecoregion (Fig. 7d) show that the optimized fluxes are either independent of (e.g. boreal forests/wooded, temperate grass/shrubs) or have a slight dependence on (e.g. boreal tundra/taiga, temperate forests/wooded) on the priors. This demonstrates that the CTDAS-Lagrange system can resolve large biases in the priors, but the magnitude of adjustment is also limited by the prescribed flux uncertainty, which is confirmed by the tests with increased flux
uncertainty (not shown). Besides this, the limited choice of data constraints also limits the ability of the system to respond to biased prior fluxes.

Finally, we note that tests of CTDAS-Lagrange in so-called OSSEs (Observing System Simulation Experiments) confirm that a near-perfect truth can be estimated with the system if pseudo-observations are created from known fluxes. In such
experiments, transport errors and systematic structural differences between truth and prior flux+BC patterns play no role, while in reality they form a well-known limiting factor to our ability to estimate surface exchange.

### 3.5 The flux adjustment method: multiplicative vs. additive

The prior/optimized fluxes using both additive and multiplicative flux adjustment methods are shown in Fig. 8. We found that major differences occur in the so-called shoulder seasons, where the flux adjustment is significant for the run with the
additive method but is negligible for the run with the multiplicative method. The multiplicative method fails to adjust the fluxes in this case because the NEE is small or even close to zero around the shoulder seasons. Larger variations in the optimized fluxes for the additive flux adjustment method are observed compared to those for the multiplicative method, due to the flexibility of the additive flux adjustment method and higher prior flux uncertainties. Note that both methods reproduce observed $CO_2$ values equally well and multiplicative scalars do not lead to worse residuals.

Fig. 9 shows the inversion results of model runs with pseudo data, further confirming the advantage of the additive method over the multiplicative method in the CTDAS-Lagrange system. The additive method recovers the seasonality better than the



multiplicative method, noticeable mainly for the period June-July. It is also clearly shown that the multiplicative method fails to derive the "truth" fluxes around the shoulder season in the fall (no difference between the prior and the truth in the spring). Besides this, the estimate of the annual net biosphere fluxes derived from the additive method is also closer to the truth than that from the multiplicative method, although the associated uncertainties are rather large.

**3.6 Sensitivity to the covariance length scale**

The sensitivity of the CTDAS-Lagrange to the covariance length scale is shown in Fig. 10. The optimized fluxes tend to reach a robust value when the covariance length scale is larger than 750-1000 km, and we note that the difference between 750 km and 1000 km is relatively small. We have tested whether including aircraft sites can reduce this length scale dependence below 1000 km, and find it can slightly alleviate the dependence but does not fully resolve that. The optimized

fluxes for the temperate North America are relatively insensitive to the covariance length scale, as this region is relatively well sampled by the dataset. We have only used some of the available observations, and different results may be found when additional data is included, e.g. from Environment Canada tower sites.

**3.7 Ensemble estimates**

From the above-described sensitivity runs, we derive an ensemble of estimates of optimized North American annual net

biosphere fluxes in 2010 (see Fig. 11). The optimized biosphere fluxes of all the runs are larger (i.e. more uptake) than their corresponding prior fluxes. Compared to other factors, the prior biosphere fluxes have the largest impact on the optimization result. The selection of model-data mismatch with 3 ppm is reasonable, judging from the observed small differences between the model runs BASE and R2 (4 ppm). We notice the R1 (2 ppm) run makes a significant difference, as it rejects much more observations than the other two cases, especially during summer time when usually larger mismatches between observations

and simulations occurred (not shown).

Comparing BASE, Q1 (decrease the additive covariance magnitude by 25%) and Q2 (increase the additive covariance magnitude by 25%), we find the prior uncertainty magnitude ascribed to biosphere fluxes impacts the result only little, with small reductions in the optimized flux when the uncertainty gets larger. In addition, we find that our system is sensitive to

the uncertainty of the boundary condition parameter Pbc1 (1 ppm unc.) and Pbc2 (3 ppm unc.), which results in the difference of flux estimates by slightly more than 0.1 PgC/yr. The choice of 2 ppm is according to the uncertainties we assessed from the statistics between these different prior BCs we used. When the two tower sites STR and WGC that are located close to the west coast of North America are excluded, we find smaller biosphere fluxes (the difference is approximately 0.15 PgC/yr), which indicates that attention should be paid to the choice of the observations.

Excluding results from B2 that we consider unrealistic due to the high data rejection rate (replaced by the B2' run), we estimate North America Carbon fluxes for the year 2010 to be between -0.92 and -1.26 PgC/yr.



### 4. Conclusions and Discussion

We have implemented a regional carbon assimilation system based on the CarbonTracker Data Assimilation Shell framework and a high-resolution Lagrangian transport model WRF-STILT. The new system, named as CTDAS-Lagrange, optimizes both biosphere fluxes and four boundary condition parameters and is computationally efficient (one year of

optimization can be performed serially within 14 hours with 8 threads on a 12-core Intel Xeon Processor E5 v2 Family computer with a processor base frequency of 2.7 GHz, once footprints are calculated and stored offline). Furthermore, we have demonstrated that the additive flux adjustment method is more flexible in optimizing NEE than the multiplicative flux adjustment method, especially in the shoulder seasons of the year.

The sensitivity test results with three different lateral boundary conditions (CT2013B, CTE2014, and an empirical curtain) indicate that CTDAS-Lagrange has the ability to largely correct for the potential biases in the lateral boundary conditions, with the boundary condition optimization absorbing up to 0.23 PgC/yr of flux adjustment that would otherwise have been made to the optimized annual net biosphere fluxes. This makes the CTDAS-Lagrange system less dependent on the choice of lateral boundary conditions than a system without boundary condition optimization or offline correction. The sensitivity tests

with two alternative biosphere fluxes (SiB3 and CT2013B-avg) and a series of modified SiBCASA fluxes with a large range of NEE show that the seasonal magnitude of the optimized fluxes is almost independent of the prior fluxes, and the optimized annual net biosphere fluxes are both consistent for SiB3 and CT2013B-avg and much less dependent on the range of the priors for the series of modified SiBCASA fluxes. This demonstrates that the CTDAS-Lagrange system is capable of resolving large biases in the prior biosphere fluxes. On the other hand, the optimized annual net biosphere fluxes per

ecoregion are less consistent, presumably due to the limited choice of observational constraints. This could also be improved by better prescribing the uncertainties of biosphere fluxes for the additive adjustment method, as the assumption of spatially and temporally uniform flux uncertainties may not be reasonable.

We derive an ensemble of estimates of the optimized annual net biosphere carbon fluxes based on a series of sensitivity tests,

which places the North American Carbon sink for the year 2010 at -0.92 to -1.26 PgC/yr, comparable to the TM5 based estimates of CarbonTracker (version CT2016, -0.91 ± 1.10 PgC/yr, data obtained from https://www.esrl.noaa.gov/gmd/ccgg/carbontracker/) and CarbonTracker Europe (version CTE2016, -0.91 ± 0.31 PgC/yr; van der Laan-Luijkx, et al. 2017). Note that much less observations have been used in CTDAS-Lagrange than those assimilated in CT2016 and CTE2016. This work is to be followed up by a multi-year inversion using more available observations in recent years, and by assimilating an

additional tracer, carbonyl sulfide, to simultaneously constrain both GPP and NEE.

In addition, the estimate of net $CO_2$ uptake for the year 2010 is reasonable compared to an ensemble of atmospheric inversions from several studies that place the North American NEE for 2000-2006 at -0.931 ± 0.670 PgC/yr (Hayes et al.,





2012), and a more recent study suggesting the North American net $CO_2$ biosphere fluxes during 2000-2009 to be -0.890 ± 0.400 PgC/yr from the RECCAP-selected TransCom3 inversions (King et al., 2015). Three atmospheric inversion studies placed North American NEE for the year 2004, which had been recognized as a climate-favorable year for uptake, from -0.953 ± 0.106 to -1.230 ± 1.120 PgC/yr (CT2011, Peters et al., 2007; Butler et al., 2010; Gourdji et al., 2012). However, we

also note that although we have a comparable estimate to the two versions of CarbonTracker at the continental scale, our estimates at ecoregion scales are different. Typically the boreal region is not well constrained in our study with eight tower sites located in the U.S, while no data from the extensive network of Environment Canada was used. To solve finer scale fluxes, the use of data from a denser observational network is desirable and could likely further reduce the chosen covariance length scale shown in Fig. 10.

Although we have accounted for the impact of possible biases in the prior lateral boundary conditions on optimized fluxes, we find that there remains room to further reduce the biases at surface sites (shown in Table 1 as posterior residuals). This could be partially because aircraft observations are sparse, and are temporally insufficient for sampling the inflows of the continent. Also, the limited number of parameters used for boundary condition adjustments could be a bottleneck; an

alternative scheme with more extensive parameterization to offer more flexibility for boundary condition adjustments could help.

Moreover, the optimization results of the CTDAS-Lagrange system depend on the quality of the forward simulations, i.e. fixed a priori fluxes and transport models. The optimization of biosphere fluxes may be influenced by observations affected

by local fossil fuel signals, which can be addressed by using high-resolution fossil fuel emissions or filtering out the observations. CO has been long studied and used as a tracer for fossil fuel emissions, and its use as a quantitative tracer suffers mainly from varying emission ratios of different sources and production by oxidation of hydrocarbons. We have used CO as a fossil fuel tracer to filter out observations that are considerably affected by fossil fuel emissions, which is expected to serve our purpose reasonably well in winter time, however, may not be appropriate on the occasions when

production of CO by oxidation of hydrocarbons can be significant in summer time. Thus the CO filtering may have been overly conservative, reducing the number of observations by up to ~7% (the average percentage at all sites) in summer time. As we use the same filtered dataset for all the model runs, and the CTDAS-Lagrange system rejects observations when the difference between simulated and observed $CO_2$ is larger than three times the assigned model-data-mismatch, the potential issue of inappropriate CO filtering in summer time is unlikely to significantly bias our results. Efforts have been made to

assimilate both $CO_2$ and $^{14}CO_2$, a tracer for recently added fossil fuel, to optimize both biosphere and fossil fuel fluxes (Basu et al., 2016; Fischer et al., 2017). To investigate the influence of transport model, we have performed tests with an alternative Lagrangian transport model, the Hybrid Single Particle Lagrangian Integrated Trajectory (HYSPLIT; Draxler and Hess 1998; Stein et al., 2015) model driven by the North American Mesoscale Forecast System meteorological fields at 12 km resolution (NAM12). Our preliminary results show that the HYSPLIT-NAM12 run yields similar biosphere uptake in





summertime, but higher respiration fluxes in wintertime. Hegarty et al, (2013) showed that the three widely used Lagrange particle dispersion models (LPDMs), HYSPLIT, STILT, and Flexible Particle (FLEXPART; Stohl et al., 2005) have comparable skills in simulating the plumes from controlled tracer release experiments when driven with identical meteorological inputs. Thus, the observed difference is mostly likely caused by the difference between WRF and NAM12.

A second reason could be attributed to the fact that the WRF domain is larger and extends much further to the north than the NAM12 files archived by NOAA Air Resources Laboratory (https://ready.arl.noaa.gov/hypub/ hysp_meteoinfo.html). Therefore, compared to the NAM12 run, the WRF-STILT run covers a larger area, and is less influenced by the potential biases in the northern boundary that is nontrivial to correct for due to the large latitudinal gradients in $CO_2$ mole fractions. A further detailed investigation into the differences between the two meteorological inputs is required to diagnose the resulting

difference in the optimized fluxes in wintertime; however, this task is beyond the scope of the development of our CTDAS-Lagrange system.

We highlight that the use of aircraft data in this study suggests a very important constraint from free tropospheric measurements to the lateral boundary conditions, which enables simultaneous optimization of boundary conditions and

biosphere fluxes. Our system is an open framework for regional atmospheric inversions that could be extended to use different atmospheric transport models, to study other trace gases, and for alternative geographic regions.

Code availability

The codes can be downloaded from https://github.com/njuhewei/CTDAS-Lagrange. The major part of the system is

programmed with Python, and the module for forward simulations is programmed with R.

Author contributions. W.H. and H.C. prepared the manuscript with contributions from all co-authors. W.H., H.C., I.R. van der V. and W.P. developed the CTDAS-Lagrange, with contributions from the other authors. A.E.A., T.H., and M.M. prepared the WRF-STILT footprints.

Competing interests. The authors declare that they have no conflict of interest.

### Acknowledgement

This research is funded by the NOAA Climate Program Office's AC4 program (award number NA13OAR4310082 and additional support for production of the NOAA CarbonTracker Lagrange footprint library), and by the National Key R&D

Program of China (2016YFA0600202). W.P. was partially funded from European Research Council Grant 649087 (ASICA). I.T. van der Laan-Luijkx is funded by a  NWO  Veni  grant  (016.Veni.171.095).     Data collection at Walnut Grove and Sutro towers  was  partially  supported  by  the California Energy Commissions's  Natural  Gas  Research  Program  and



the California Air Resources Board at Lawrence Berkeley National Laboratory under U.S. Department of Energy Contract No. DE-AC02-05CH11231.We are grateful to Ian Baker for providing the SiB3 fluxes, to Andy Jacobson for his useful comments on the manuscript, and for the IT support at the Center for Isotope Research of the University of Groningen.

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





**Table 1.** Summary of assimilated PFP flask data from the NOAA ESRL North American tall tower and aircraft sampling program in 2010. We have selected 12 aircraft sites for this study. Observations (after data filtering, see section 2.1.3) are flagged when the difference between simulated and observed values is larger than three times the prescribed model-data-mismatch for each site. The bias indicates the mean difference between model forecast and observations.

| Code | Name | Lat, Lon, Elev | Number of obs. (rejected) | Model-data mismatch (ppm) | Inn. $\|^2$ | Prior residuals (ppm) | Posterior residuals (ppm) |
|---|---|---|---|---|---|---|---|
| *surface sites* | | | | | | | |
| AMT | Argyle, Maine, United States | 45° 2'N, 68°41'W, 53 masl | 347 (9) | 3 | +0.51 | +1.00± 3.48 | +0.05± 1.52 |
| BAO | Erie, Colorado, United States | 40° 3'N, 105° 0'W, 1584 masl | 396 (17) | 3 | +0.44 | -0.77± 3.90 | +0.05± 2.33 |
| LEF | Park Falls, Wisconsin, United States | 45°57'N, 90°16'W, 472 masl | 409 (11) | 3 | +0.49 | +0.88±3.88 | -0.09± 1.55 |
| SCT | Beech Island, South Carolina, United States | 33°24'N, 81°50'W, 115 masl | 437 (15) | 3 | +0.60 | +1.03±3.85 | -0.21± 1.82 |
| STR | Sutro Tower, San Francisco, California, United States | 37°45'N, 122°27'W, 254 masl | 215 (6) | 3 | +0.91 | +2.00±3.86 | +0.60± 2.14 |
| WBI | West Branch, Iowa, United States | 41°43'N, 91°21'W, 241 masl | 472 (36) | 3 | +0.69 | +0.08±4.99 | -0.16± 1.63 |
| WGC | Walnut Grove, California, United States | 38°16'N, 121°29'W, 0 masl | 393 (71) | 3 | +0.87 | +0.52±7.66 | -0.08± 3.48 |
| WKT | Moody, Texas, United States | 31°19'N, 97°20'W, 251 masl | 353 (10) | 3 | +0.50 | +0.68±3.53 | +0.11± 1.63 |
| *aircraft sites* | | | | | | | |
| CAR | Briggsdale, Colorado, United States | 40°22'N, 104°18'W, 1740 masl | 139 (1) | 1 | +0.37 | +0.06±0.80 | -0.02± 0.76 |
| CMA | Cape May, New Jersey, United States | 38°50'N, 74°19'W, 0 masl | 141 (3) | 1 | +0.92 | +0.13±1.12 | +0.11±0.97 |
| DND | Dahlen, North Dakota, United States | 47°30'N, 99°14'W, 472 masl | 50 (5) | 1 | +0. 78 | +0. 35±1.56 | -0.00±0.89 |
| ESP | Estevan Point, British Columbia, Canada | 49°23'N, 126°33'W, 7 masl | 146 (2) | 1 | +0.96 | -0. 17±1.14 | -0. 05±0.97 |
| ETL | East Trout Lake, Saskatchewan, Canada | 54°21'N, 104°59'W, 492 masl | 126 (23) | 1 | +1.20 | +0.80±2.01 | +0.19± 1.02 |
| LEF | Park Falls, Wisconsin, United States | 45°57'N, 90°16'W, 472 masl | 37 (4) | 1 | +0.80 | +0.53±1.72 | +0.15± 0.87 |
| NHA | Worcester, Massachusetts, United States | 42°57'N, 70°38'W, 0 masl | 150 (17) | 1 | +1.69 | +0.61±2.51 | +0.05± 0.98 |
| PFA | Poker Flat, Alaska, United States | 65° 4'N, 147°17'W, 210 masl | 95 (4) | 1 | +1.38 | +0. 15±1.54 | +0.11± 1.07 |
| SCA | Charleston, South Carolina, United states | 32°46'N, 79°33'W, 0 masl | 130 (0) | 1 | +0.51 | +0. 16±0.72 | +0.22± 0.63 |
| SGP | Southern Great Plains, Oklahoma, United states | 36°36'N, 97°29'W, 314 masl | 88 (1) | 1 | +0.91 | +2. 00±3.86 | +0.60±2.14 |
| TGC | Sinton, Texas, United States | 27°44'N, 96°52'W, 0 masl | 124 (6) | 1 | +0.66 | -0.01±0.82 | -0.01± 0.79 |
| THD | Trinidad Head, California, United States | 41° 3'N, 124° 9'W, 107 masl | 31 (1) | 1 | +1.55 | +1.02±1.13 | +0.70± 0.91 |





**Table 2.** Summary of the base and sensitivity runs using CTDAS-Lagrange.

| | Covariance length scale (unit: km) | Prior biosphere fluxes | Prior boundary conditions | Uncertainty of boundary conditions | Uncertainty of biosphere fluxes | Model-data mismatch* (unit: ppm) | Observations used | Transport model |
|---|---|---|---|---|---|---|---|---|
| **Base** | 750 | SiBCASA | CT2013B | 2.0 | - | 3 | All | WRF-STILT |
| **CL1** | 300 | SiBCASA | CT2013B | 2.0 | - | 3 | All | WRF-STILT |
| **CL2** | 500 | SiBCASA | CT2013B | 2.0 | - | 3 | All | WRF-STILT |
| **CL3** | 1000 | SiBCASA | CT2013B | 2.0 | - | 3 | All | WRF-STILT |
| **CL4** | 1250 | SiBCASA | CT2013B | 2.0 | - | 3 | All | WRF-STILT |
| **B1** | 750 | SiB3 | CT2013B | 2.0 | - | 3 | All | WRF-STILT |
| **B2** | 750 | CT2013B | CT2013B | 2.0 | - | 3 | All | WRF-STILT |
| **B2'** | 750 | CT2013B-avg | CT2013B | 2.0 | - | 3 | All | WRF-STILT |
| **BX2** | 750 | SiBCASA | CT2013B | 2.0 | - | 3 | All | WRF-STILT |
| **BC1** | 750 | SiBCASA | CTE2014 | 2.0 | - | 3 | All | WRF-STILT |
| **BC2** | 750 | SiBCASA | EMP | 2.0 | - | 3 | All | WRF-STILT |
| **Pbc1** | 750 | SiBCASA | CT2013B | 1.0 | - | 3 | All | WRF-STILT |
| **Pbc2** | 750 | SiBCASA | CT2013B | 3.0 | - | 3 | All | WRF-STILT |
| **Q1** | 750 | SiBCASA | CT2013B | 2.0 | 50% of default | 3 | All | WRF-STILT |
| **Q2** | 750 | SiBCASA | CT2013B | 2.0 | 150% of default | 3 | All | WRF-STILT |
| **R1** | 750 | SiBCASA | CT2013B | 2.0 | - | 2 | All | WRF-STILT |
| **R2** | 750 | SiBCASA | CT2013B | 2.0 | - | 4 | All | WRF-STILT |
| **Obs** | 750 | SiBCASA | CT2013B | 2.0 | - | 3 | excl. STR & WGC | WRF-STILT |

\* For the tower site. A model-data mismatch of 1 ppm of is used for aircraft observations in all simulations.





**Table 3.** Contribution of lateral transport to simulated $CO_2$ mole fractions at the tall tower network for three sets of lateral boundary conditions. The mean differences (in ppm) between CTE2014, EMP, and CT2013B boundary conditions are calculated for 2010 and summer (JJA) respectively. The standard deviation (1-sigma) of the differences for each tower site is given in the parenthesis.

| Site | CTE2014 minus CT2013B | | EMP minus CT2013B | |
|------|------------|------------|------------|------------|
|      | annual | summer | annual | summer |
| AMT | 0.39 (±0.43) | 0.49 (±0.56) | 0.04 (±1.27) | -1.19 (±1.33) |
| BAO | 0.25 (±0.34) | -0.11 (±0.35) | -0.24 (±1.26) | -1.70 (±1.54) |
| LEF | 0.36 (±0.38) | 0.00 (±0.32) | 0.05 (±1.36) | -1.07 (±1.76) |
| SCT | 0.29 (±0.37) | 0.01 (±0.40) | -0.07 (±1.19) | -0.50 (±1.16) |
| STR | 0.35 (±0.52) | 0.30 (±0.19) | -0.21 (±1.58) | 0.66 (±0.78) |
| WBI | 0.39 (±0.41) | 0.16 (±0.29) | -0.03 (±1.21) | -0.28 (±1.10) |
| WGC | 0.35 (±0.39) | -0.00 (±0.34) | -0.06 (±1.30) | -0.93 (±1.80) |
| WKT | 0.31 (±0.38) | 0.16 (±0.61) | 0.14 (±1.08) | -0.66 (±1.67) |



**Table 4.** Comparison of the optimized annual net biosphere fluxes (PgC/yr) and the adjustment of $CO_2$ boundary conditions (ppm) using different prior lateral boundary condition products (CT2013B, CTE2014, and EMP) and optimization techniques ("Flux only" or "Flux + BC"). The annual net biosphere flux difference is calculated from the "Flux + BC" optimization minus "Flux only" optimization.

| Total flux | Base (CT2013B) | BC1 (CTE2014) | BC2 (EMP) |
|---|---|---|---|
| **"Flux only" optimization (PgC/yr)** | -1.10 (±1.75) | -1.49 (±1.75) | -1.09 (±1.75) |
| **"Flux + BC" optimization (PgC/yr)** | -1.08 (±1.74) | -1.26 (±1.74) | -1.09 (±1.73) |
| **Flux difference (PgC/yr)** | +0.02 | +0.23 | -0.00 |
| **BC adjustment (ppm)** | -0.17 to 0.14 | -0.18 to 0.16 | -0.08 to 0.09 |



**Table 5.** Optimized annual net biosphere fluxes (PgC/yr) and $CO_2$ boundary condition adjustments (ppm) using four prior biosphere flux products (SiBCASA, SiB3, CT2013B, and CT2013B-avg).

| Total flux | Base (SiBCASA) | B1 (SiB3) | B2 (CT2013B) | B2' (CT2013B-avg) |
|---|---|---|---|---|
| Prior (PgC/yr) | -0.51 (±4.66) | -0.57 (±4.66) | -0.43 (±4.66) | -0.44 (±4.66) |
| Optimized (PgC/yr) | -1.08 (±1.74) | -1.01(±1.74) | -0.65 (±1.75) | -1.20 (±1.74) |
| Flux adjustment (PgC/yr) | -0.57 | -0.44 | -0.21 | -0.76 |
| BC adjustment (ppm) | -0.17 to 0.14 | -0.17 to 0.14 | -0.17 to 0.14 | -0.17 to 0.14 |



**Table 6.** Sensitivity runs with a variety of prior biosphere fluxes ranging from +0.43 to -2.06 PgC/yr for North America. The prior biosphere fluxes were derived by scaling up or down the SiBCASA respiration estimate while maintaining the same GPP estimate. The flux adjustment is calculated from the optimized estimate minus the prior estimate.

|  | BX1 | BX2 | Base | BX3 | BX4 | BX5 |
|---|---|---|---|---|---|---|
| **Prior (PgC/yr)** | +0.43 (±4.66) | -0.06 (±4.66) | -0.51 (±4.66) | -0.97 (±4.66) | -1.44 (±4.66) | -2.06 (±4.66) |
| **Optimized (PgC/yr)** | -0.90 (±1.74) | -1.01 (±1.74) | -1.08 (±1.74) | -1.21(±1.74) | -1.32 (±1.74) | -1.45 (±1.74) |
| **Flux adjustment (PgC/yr)** | -1.33 | -0.95 | -0.57 | -0.24 | +0.12 | +0.61 |





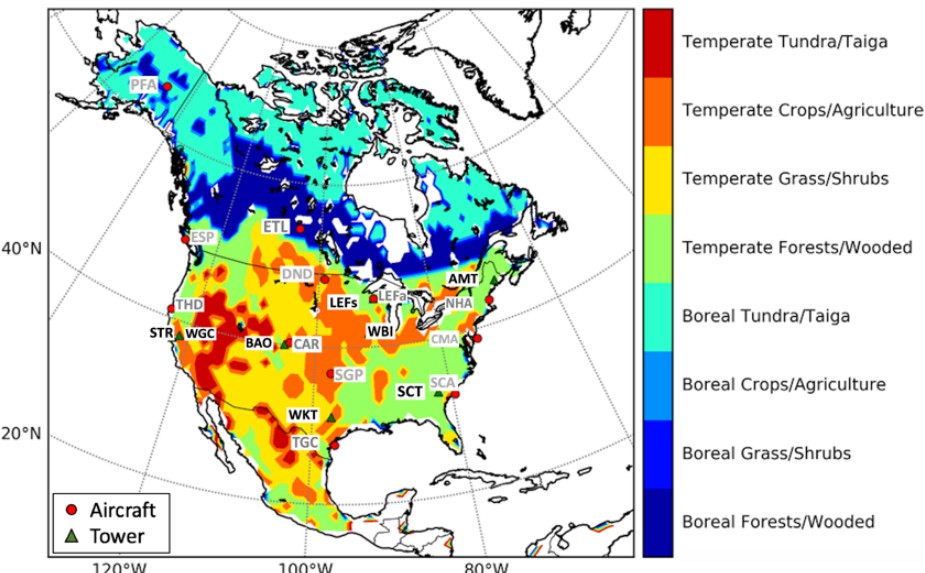

**Figure 1.** The model domain is shown together with the $CO_2$ observational sites from NOAA's Global Greenhouse Gas Reference Network and the aggregated Olson ecosystem types. Eight tall tower sites are highlighted by green triangles with black site code labels, and twelve selected aircraft sites are highlighted by red dots with gray site code labels.



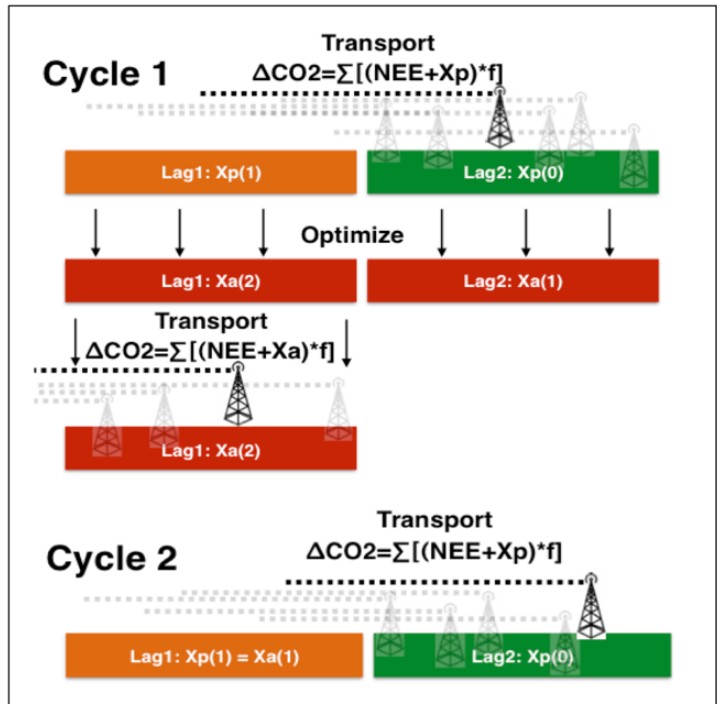

**Figure 2.** The time stepping flow of the Ensemble Kalman Smoother filter used in CTDAS-Lagrange. The $Xp(n)$ and $Xa(n)$ represent the prior and optimized state vector of a 20-day period shown as two colored bars from left to right. Parameter n denotes the number of times the state vector has been updated. Each of the colored bars represents a 'lag' of 10 days. In the *transport step* we calculate the $CO_2$ mole fraction variations ($\Delta CO_2$) for each measurement (shown as a tower) by convolving the net biosphere flux *NEE + Xp/Xa* with footprints *f* (shown as a dashed line). Cycle 1 starts by introducing new set of observations and fluxes at the front of the filter in lag 2. This part of the state vector has not been optimized before (green color, n=0). At the rear end of the filter, in lag 1, the state vector has been optimized once in the previous cycle (orange color, n=1). To estimate $\Delta CO_2$ for each observation requires convolving footprints with 10-day 3-hourly *NEE + Xp*. Optimization is done on all 20 days (2 lags of the filter) to find the optimal values in the entire state vector. The state vector of lag 1 is done and will not change again (red color, n=2). This new optimized state vector $Xa(2)$ is used to calculate the final $\Delta CO_2$ in lag 1 (final *transport step*). Cycle 2 starts by introducing a new set of observations and fluxes at the front of the filter in lag 2. The analyzed state vector $Xa(1)$ in lag 2 of Cycle 1 becomes the new prior state vector $Xp(1)$ in lag 1 of Cycle 2.





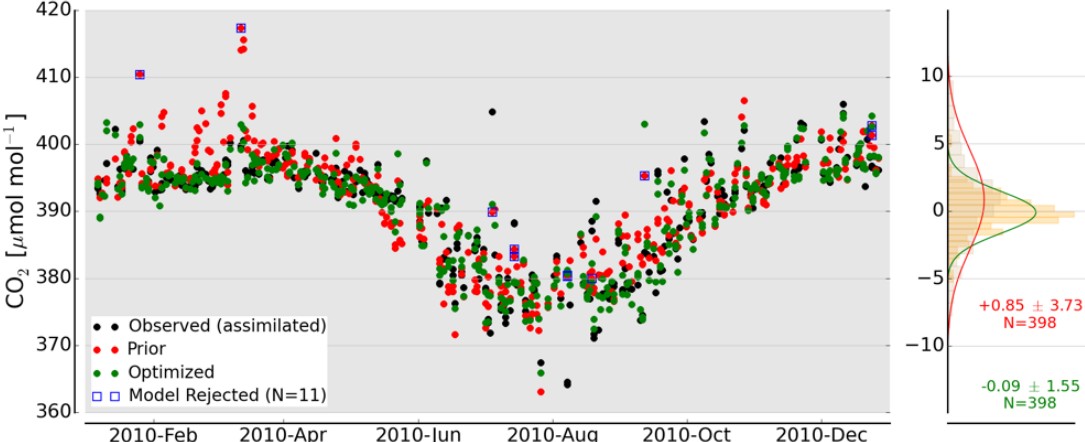

**Figure 3.** Simulated $CO_2$ (prior in red and optimized in green) and observed $CO_2$ (black) for the Park Falls, Wisconsin tower site (LEF) for the year 2010. The blue squares (11 out of 409 samples) are rejected samples because the difference between simulated and observed $CO_2$ is larger than three times the assigned model-data-mismatch of 3 ppm for tower sites. The distribution of both prior and posterior residuals is presented on the right side. After optimization we observe a strong reduction of the $CO_2$ mean bias and $1\sigma$ standard deviation from $+0.85 \pm 3.73$ ppm to $+0.09 \pm 1.55$ ppm. Note that the prior residual distribution is calculated without the rejected observations, which explains the slightly different statistics in comparison to the data presented in Table 1.





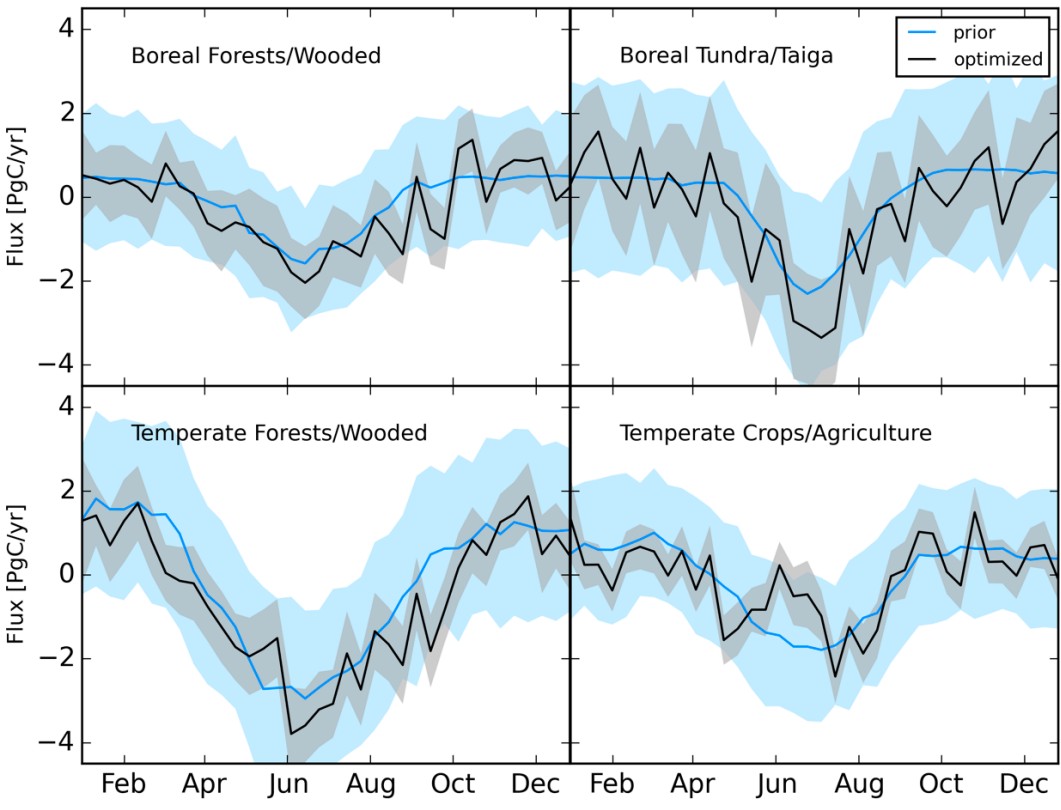

**Figure 4.** The seasonal cycle (10-day resolution) of the net $CO_2$ biosphere flux (PgC/yr) of four aggregated Olson ecoregions in North America for the year 2010. The prior biosphere flux (SiBCASA) and its uncertainty are displayed in blue, and the optimized biosphere flux and uncertainty in black.





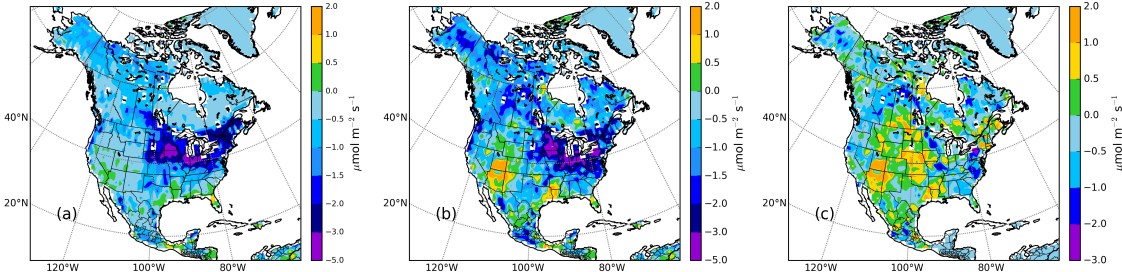

**Figure 5.** Mean prior (a) and optimized (b) net biosphere fluxes, and the mean adjustment (optimized minus prior) (c) for summertime (June-July-August). Note that the color scale used in (c) is different from the one used in (a) and (b).





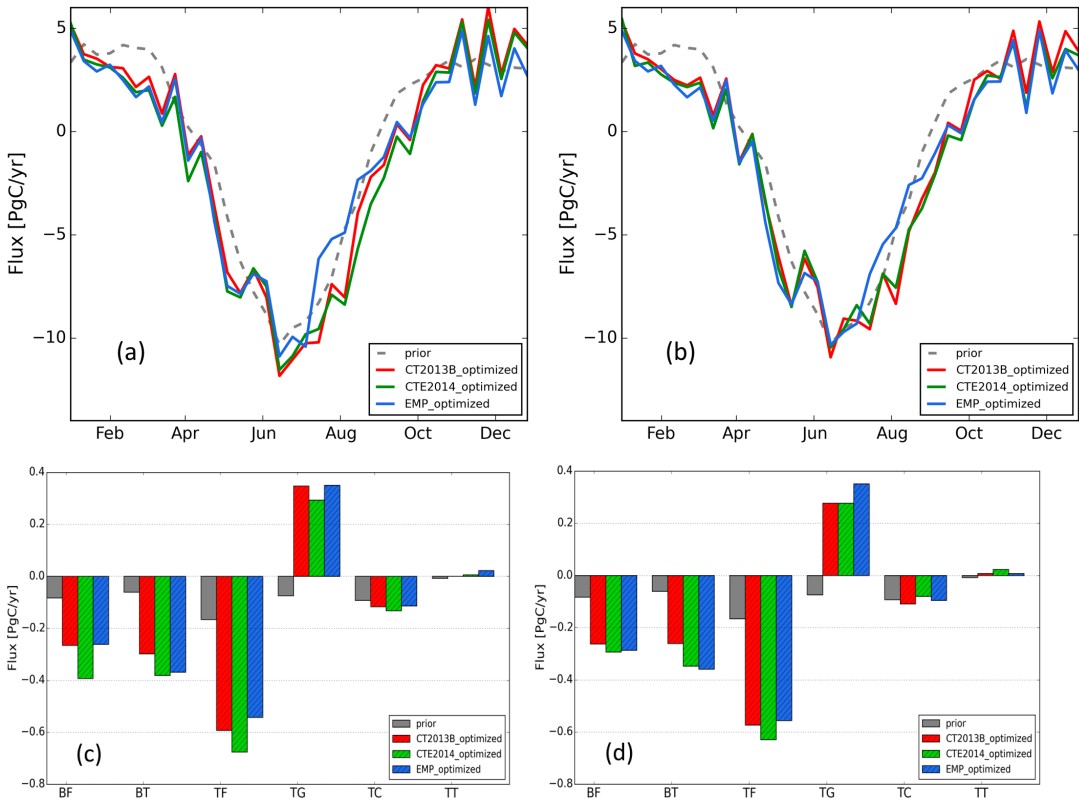

**Figure 6.** Mean optimized net biosphere fluxes (PgC/yr) for the runs with different prior boundary conditions: CT2013B, CTE2014, and
5 EMP. The time series of optimized fluxes are presented for the "Flux only" inversion (a) and for the "Flux + BC" inversion (b),
respectively; the annual net biosphere fluxes over the aggregated Olson ecosystem types are shown for the "Flux only" inversion (c) and
for the "Flux + BC" inversion (d), respectively. Note that in figures (c) and (d) the first letters "B" and "T" of the x-axis labels are short for
"Boreal" and "Temperate", respectively; the second letters "F","G","C" and "T" are short for ecosystem types "Forests/Wooded", "Grass/
Shrubs", "Crops/Agriculture ", and "Tundra /Taiga", respectively.





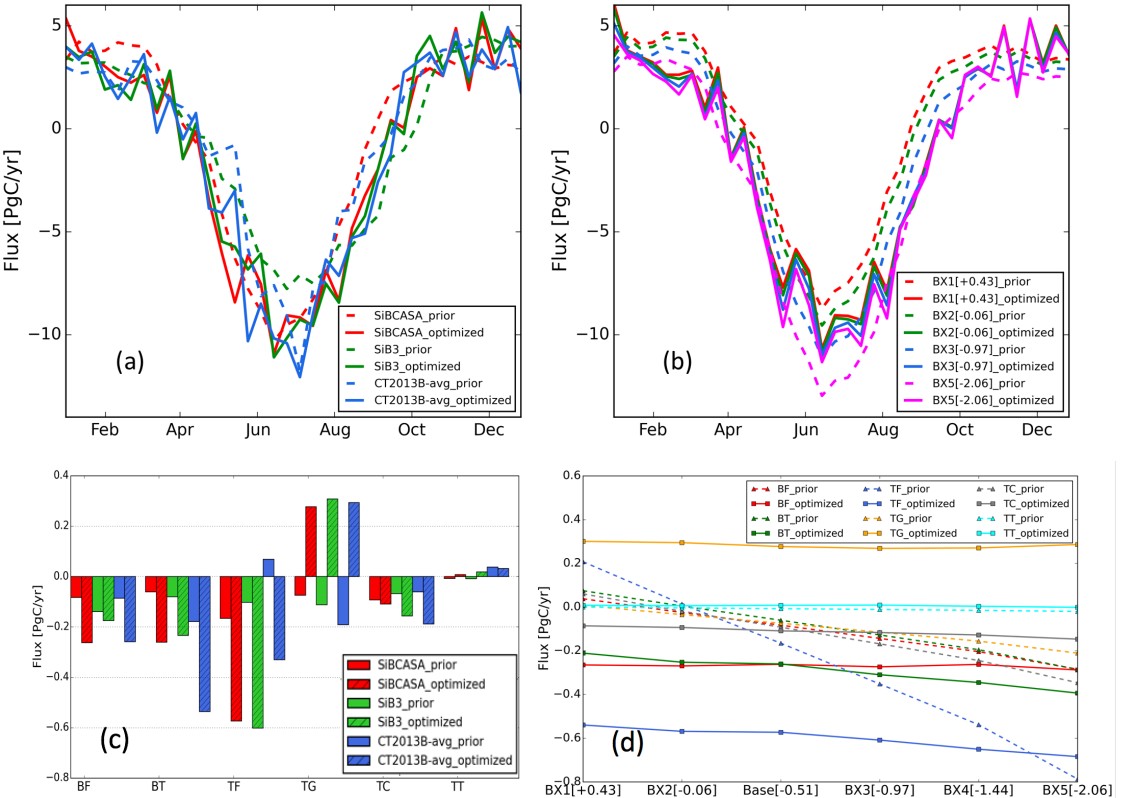

**Figure 7.** Prior and optimized annual net biosphere fluxes (PgC/yr) for North America for the runs using different prior biosphere model products: (a) SiBCASA, SiB3, CT2013B-avg, and (b) a series of modified SiBCASA fluxes derived from scaling up and down respiration fluxes. The time series of the optimized fluxes for both cases are presented in (a) and (b), respectively, and the annual net biosphere fluxes aggregated per ecoregion are accordingly presented in (c) and (d), respectively. The tests with prior fluxes +0.43, -0.06, -0.97, -1.44 and -2.06 PgC/yr are labeled with BX1, BX2, BX3, BX4 and BX5, respectively.

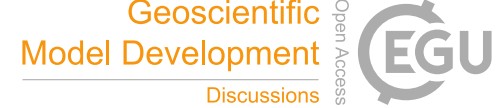

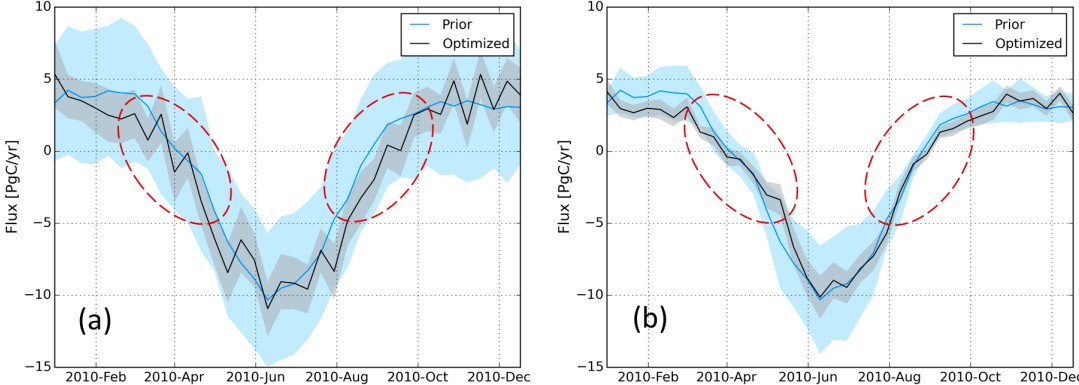

**Figure 8.** Comparison between two flux optimization methods: the additive method (a) gives significant different optimized fluxes, highlighted by the red dashed circles, in contrast to the multiplicative method (b). The additive method seems more flexible to adjust fluxes in case net carbon exchange is small or even close to zero around the shoulder seasons (spring/autumn).





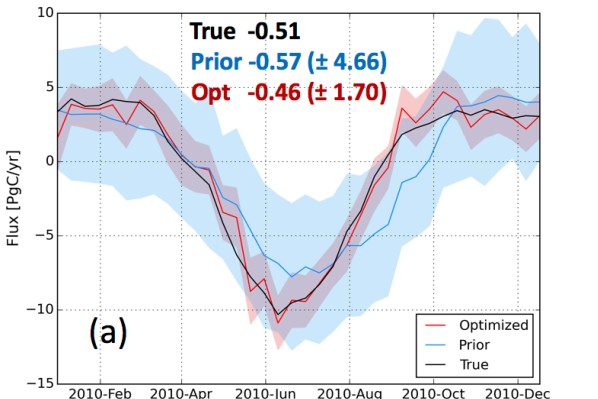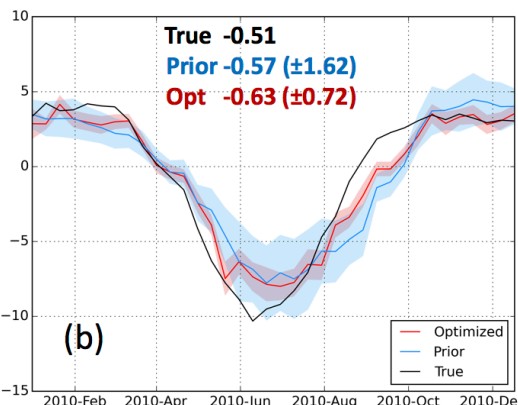

**Figure 9.** Comparison of the performance of inversions with pseudo data using the two flux optimization methods: (a) the additive method and (b) the multiplicative method. The "truth" fluxes are generated in a forward simulation using biosphere fluxes from SiBCASA. The same SiB3 fluxes are used as a priori for both runs. The annual net biosphere fluxes of the truth, prior, and optimized are given in the legend, with the unit of PgC/yr.





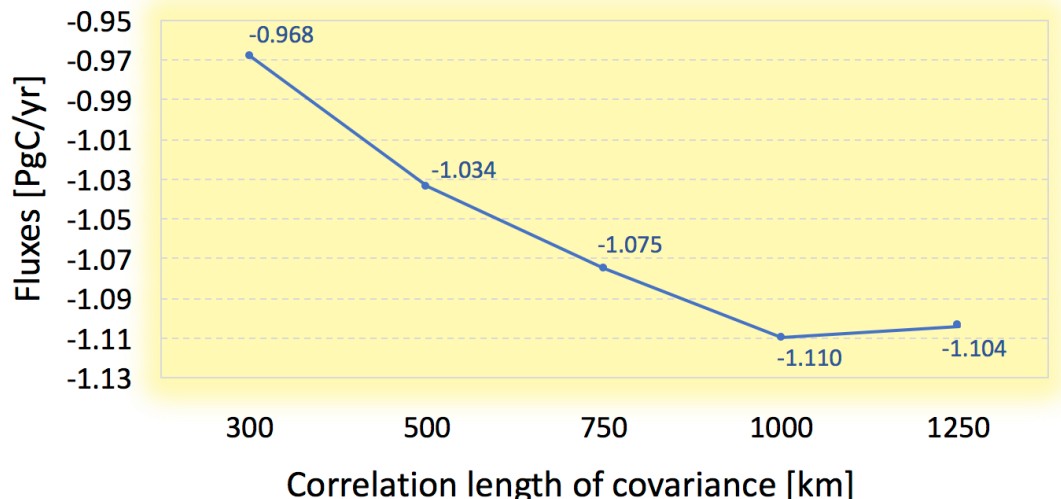

**Figure 10.** Sensitivity of the optimized annual net biosphere fluxes (PgC/yr) as a function of the chosen covariance length scale (km). The optimized fluxes tend to converge to -1.1 PgC/yr when the length scale is larger than 750 km.





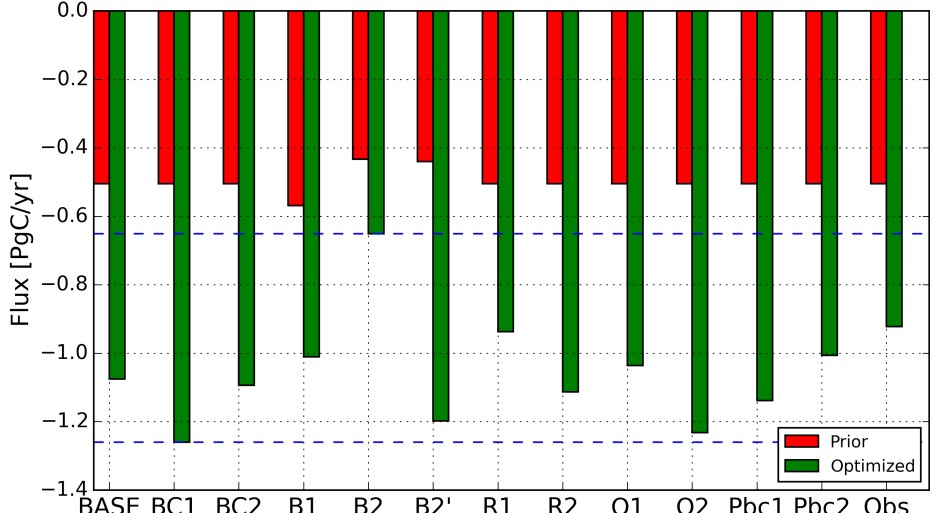

**Figure 11.** North American 2010 annual net biosphere fluxes (PgC/yr) estimated from an ensemble of CTDAS-Lagrange runs with different prior biosphere fluxes, different $CO_2$ boundary conditions, and model setup choices. The dashed blue lines refer to the range of the ensemble estimates -0.92 to -1.26 PgC/yr excluding the B2 run. See Table 2 for an overview of all experiments.

