# Peer review of "CTDAS-Lagrange v1.0: A high-resolution data assimilation system for regional carbon dioxide observations"

_Geoscientific Model Development, 2017_

## Short Comment (SC1) · 13 Dec 2017

GMD is encouraging authors to provide a persistent access to the exact version of the source code used for the model version presented in the paper. As explained in https://www.geoscientific-model-development.net/about/manuscript_types.html the preferred reference to this release is through the use of a DOI which then can be cited in the paper. For projects in GitHub a DOI for a released code version can easily created using Zenodo, see https://guides.github.com/activities/citable-code/ for details. Please note that in the code accessibility section you can still point the reader to the GitHub repository for the newest version even if you use a DOI for the relevant release.

[Figure]

Lutz Gross GMD Executive Editor

---

## Referee Comment (RC1) · Anonymous Referee #1 · 18 Jan 2018

The manuscript describes a regional scale inversion system or data assimilation system to derive biosphere-atmosphere fluxes of CO2 for North America. The regional system is largely based on CarbonTracker, but the paper describes a number of experiments to specifically assess the uncertainties in the regional flux estimates. The paper is well written, however a few aspects need to be addressed before I can recommend accepting the manuscript for publication.

Main comments:

1. The derived posterior fluxes are extremely variable at sub-seasonal time scales; in P12 L30 it is mentioned that these fluctuations may be related to "artifacts that are

caused by the sparseness of the observations". It should be investigated as to whether there might be some temporal averaging or aggregation required to reduce the noise.

2. Why are the footprints aggregated to 1x1 degrees, given the 10x10 km spatial resolution of WRF resolution, and given the spatial variability of fluxes (including anthropogenic emissions) combined with rather small footprint areas within the proximity of the atmospheric observations? Has any sensitivity analysis been performed to assess the impact of using these rather coarse footprints? Furthermore, it should be clarified if SiBCASA uses only dominant vegetation types at the 1x1 degree resolution, or whether a tile approach is used to also include other vegetation types within a given grid cell?

3. The shortest length scales used for the prior error covariance are rather large compared to those suggested e.g. by Chevallier et al. (2012) or by Kountouris et al. (2015). Given that the change in the annual net biosphere fluxes with correlation length scale seems to become larger with shorter length scales (Fig. 10), also short correlation scales should be investigated. Furthermore, fig. 10 should include the prior flux and the prior and posterior uncertainties (which might change with correlation scale depending on the setup).

Detailed comments:

P4 L7: Table 1 lists 12 rather than six PFP aircraft sites, also P4 L28 mentions 12 sites

P6 L18: In equation (1) beta_i, W_i and S should also have a dependence on observation time $t_r$ and location $X_r$.

P6 L23-25: This is unclear. Particles leaving the domain below 3000 m are not considered, does that mean they get a zero boundary condition value for the mole fraction? Even though the influence from surface fluxes is strong, the lateral boundary condition for those should be quite different from zero. What about those particles that did not leave the domain within the 10 days?

P7 L24: Why use 3 hourly mean fluxes, as footprints are available at hourly time steps? Was the associated aggregation error quantified? Similar with fossil fuel emissions (P8 L5)

P8 L26: "multi-model prior suite of inversion" this is unclear, may be reformulate?

P9 L12: Lateral boundary condition EMP: references only describe LBCs for gases other than CO2. A more detailed description of how EMP CO2 fields were derived is needed. How does the EMP boundary condition differ from the one described in Gerbig et al., 2003?

P10 L18, Table 2: Cases BX1-BX5 should be included in Table 2 (there only BX2 is included, but the values for the different columns are identical to those for the Base run). Also, the run B2' (included in Table 2) should be described in the text. Similarly, the multiplicative flux adjustment run should be included in Table 2.

Table 4: the values in brackets need to be explained in the table caption. Also "BC adjustment" should be given as a mean and a range, and the range should be explained in the caption.

Table 5: the values in brackets need to be explained in the table caption, I assume those are the uncertainties.

P14 L20: what is meant by "consistent"? Given the uncertainties all flux estimates are statistically indistinguishable.

P14 L23-29: unclear, how the averaging was done. Did the prior fluxes in the CT2013B-avg case have any diurnal variations? The reason for using 10-day averages is also not clear to me. CTDAS-Lagrange uses 10-day backward calculations, but the resulting footprint values change strongly with backward time (time before measurement time).

P15 L20: OSSEs can be set up in very different ways, allowing also for differences in transport (using different transport models) or structural differences in biospheric fluxes. This should be reformulated.

[Figure]

P15 L23, Fig. 8, Fig. 9: Why are the prior uncertainties in the additive method so much larger than in the multiplicative method? For a clear comparison between the two methods, the underlying uncertainties should be better matched. SCALE at which matching is needed?

P16 L22: This seems inconsistent with the numbers given in Table 2 and with the description in P11 L22

References: Chevallier, F., Wang, T., Ciais, P., Maignan, F., Bocquet, M., Altaf Arain, M., Cescatti, A., CHEN, J., Dolman, A. J., Law, B. E., Margolis, H. A., Montagnani, L. and Moors, E. J.: What eddy-covariance measurements tell us about prior land flux errors in $CO_2$-flux inversion schemes, Global Biogeochem. Cycles, 26(1), doi:10.1029/2010GB003974, 2012.

Kountouris, P., Gerbig, C., Totsche, K. U., Dolman, A. J., Meesters, A. G. C. A., Broquet, G., Maignan, F., Gioli, B., Montagnani, L. and Helfter, C.: An objective prior error quantification for regional atmospheric inverse applications, Biogeosciences, 12(24), 7403–7421, doi:10.5194/bg-12-7403-2015, 2015.

---

## Referee Comment (RC2) · Anonymous Referee #2 · 2 Apr 2018

Review of the paper entitled "CTDAS-Lagrange v1.0: A high-resolution data assimilation system for regional carbon dioxide observations" by He et al.

General comment:

The regional assimilation system presented here is the first semi-operational atmospheric inversion of carbon fluxes at the mesoscale. Compared to previous inversion systems, CTDAS-Lagrange has been significantly improved for high-resolution problems thanks to a state-of-the-art atmospheric model and a comprehensive optimization framework including the problem of boundary conditions. The inversion framework is similar to CarbonTracker but has been adapted to the optimization of a pixel-based

[Discussion paper]

[Figure]

state vector. The sensitivity experiments performed here shows that the system is capable of reproducing the observed variability in CO2 mixing ratios across North America. The continental fluxes are consistent with global-scale inversions, and the robustness of the optimized surface fluxes to assumptions made in the inversion has been tested carefully. Therefore, we recommend this paper for publication after considering the general and specific comments listed here.

Boundary Conditions: Three products have been used here to describe the CO2 mixing ratios coming from outside the simulation domain. The results show that two of them (i.e. CT2013B and EMP) produce very similar results with nearly identical posterior fluxes. However, the third one (i.e. CTE2014) produces an offset in the late growing season which is not corrected for after inversion. The major concern here is related to the representativity of the three products. The fact that one of them remains significantly different despite the optimization process suggests that the uncertainties in the boundary conditions are not removed by the assimilation of data. Hence, the ensemble has to be representative of the actual uncertainties in boundary conditions to be properly propagated into the flux uncertainties. The authors have not clearly demonstrated that these three products represent the actual errors coming the boundary conditions. This study needs to demonstrate the value of the sensitivity tests and address more carefully the actual error propagation into the posterior fluxes and their uncertainties.

Multiplicative versus additive methods: Because the prior error variances differ significantly between the two experiments, the differences between the two methods depend on the prior error variance more than the actual method to invert the fluxes. The authors need to perform another simulation with similar prior errors to produce convincing evidences that the method used is the fundamental problem.

Correlation length scale in prior errors: In a system with a fairly small degree of freedom such as CTDAS, the spatial attribution of flux corrections may still be sensitive to the definition of the prior errors but the total flux is likely to remain unchanged. The convergence of the system using different length scales can be an artefact due to the degree

of freedom but the spatial distribution of flux corrections may vary across the domain. The authors need to include maps of the flux corrections for the different length scales which will show the actual impact of prior error assumptions.

Technical comments:

P2-L10: add a reference.

P2-L11 and L12: Add references to previous studies.

P2-L14: This is true for global inversions but domain-limited inversions have opened boundaries. Refine this statement.

P2-L22: Feng et al., 2016 is not an inversion study.

P2-L25: Eulerian models are often a pre-requisite to Lagrangian models, like in CT-DAS. You may refer here to the model used in the assimilation framework, typically ensemble-based methods based on Eulerian models and analytical methods with a linearized adjoint model. This statement is unclear for column-based measurements. Refine the statement.

P2-L32: Peylin et al. (2005) discuss the importance of initial conditions in a global inversion. The problem is different for the lateral flow in a domain-limited inversion which does not decrease over time. Another citation is needed here.

Figure 1: The interpolation for land cover creates artificial zones between land cover types, with halos around temperate Crops / Agriculture for example. The native resolution of the land cover plotted for each grid cell is better suited for land cover maps. Replace contours by actual grid cell colors.

P4-L21: Indicate the time period over which the flasks have been sampled with the new protocol. How did you treat data with potential biases?

P5-L5: Have you considered the possible impact of surface fluxes when sampling at 3km and above? Some of the aircraft sampling locations could be impacted by vertical transport of surface fluxes. Could you use CO to detect high-altitude surface flux influence?

P6-L3: Add a reference to previous studies.

P6-L7: The description of the WRF model configuration is critical for future users of the footprints. Provide a complete description of the model, as well as the simulation domain (map projection, . . .).

P6-L21: You assume here that the prior BC errors are dominant over 10 days. Is that consistent with model-mismatches? Synoptic systems are more likely to be the main source of errors in the inflow. Justify the 10-day optimization window for boundary conditions.

P6-L24: You assume here that the errors in the BC's are identical below and above 3000m. This assumption seems very unlikely as the modeled values near the surface will differ significantly from the modeled mixing ratios in the Free Troposphere. An evaluation of the model-data mismatches compared to the altitude would help infer the actual vertical structures in BC errors. P8-L2: Which observations are assimilated here? Daily afternoon averages?

P11-L18: Justify the removal of outliers here. The factor of 3 applied to the MDM is arbitrary. Any physical reason behind this?

P12-L25: The reduction in uncertainty is a direct consequence of the optimization process but does not mean that the actual errors are reduced. Similar to the posterior CO2 mixing ratio mismatches, the optimization was designed to reduce it, except that fluxes have no guarantee that the reduction is real. Clarify in the paragraph.

Figure 4: The aggregation of large ecoregions over the entire continent makes the interpretation of the results very difficult. The ecoregions are too wide to make sense of the seasonal cycles that correspond to multiple regions and climates. The separation of the results into smaller regions is needed here. For example, southern and northern

regions for crops would provide a better sense of what the seasonal should look like. This figure needs to represent the ecosystems defined in climate zones.

P13-L26: - Except for CTE2014, the adjustment to the boundary conditions converges to identical posterior flux values for CT2013B and EMP. These two products are fundamentally different, but the correction to the BC's are significantly different, which suggests that the inversion is able to correct and hence reduce the initial differences between the two. However, CTE2014 produces a different result, which would suggest that the BC errors are too small, or that the lack of data limits the ability of CTDAS to reduce the BC mismatches. Provide more elements here to understand what caused the final disagreement between CTE2014 and the two other BC's. - Even for the CTE2014 case, the limited impact on the optimized surface fluxes is also surprising. Less than 0.25 PgC/yr is small compared to the large posterior uncertainties (about 1.7 PgC/yr). Is your ensemble truly representative of the BC errors? The BC's need to be illustrated here with the differences across the different products before inversion. Is the simplification into four boundaries too limiting in the optimization?

Section 3.5: The flux adjustment compares two different prior errors with significantly different values in the shoulder season. Therefore, the methods disagree mainly when the prior errors are too small to let the inversion adjusts the fluxes. The comparison is presented based on the nature of the correction, but it turns out to illustrate the importance of larger errors when the NEE value is small, and hence the scaled prior variances are small as well. Here, you illustrated the dependence to prior error variances instead of a method-dependence.

Section 3.6: This exercise requires a map of the posterior corrections. Assuming that the number of towers is sufficient to constrain most of North America, the total carbon budget will remain fairly similar. It also confirms that your degree of freedom is small, as described earlier in the paper. In this case, changing the correlation length scale will not lead to any significant changes because the inversion is over-constrained. Only biases will alter the results, such as systematic offsets in the BC. A figure illustrating

the spatial distribution of flux corrections is needed here.

**[GMDD](https://doi.org)**

Interactive
comment

---

## Author Comment (AC1) · 25 May 2018

We have created a link with a DOI number https://doi.org/10.5281/zenodo.1234231 and add it to the code accessibility section of the manuscritpt.

---

## Author Comment (AC2) · 25 May 2018

We thank two anonymous reviewers for their valuable comments. We have used them to improve the manuscript. Our point-to-point responses to the reviewers' comments are provided below.

**Reply to Referee #1**

The manuscript describes a regional scale inversion system or data assimilation system to derive biosphere-atmosphere fluxes of CO2 for North America. The regional system is largely based on CarbonTracker, but the paper describes a number of experiments to specifically assess the uncertainties in the regional flux estimates. The paper is well written, however a few aspects need to be addressed before I can recommend accepting the manuscript for publication.

Authors: We thank the referee for the appreciation of our work for regional flux estimates. We have improved the manuscript according to the comments below.

**Main comments:**

1. The derived posterior fluxes are extremely variable at sub-seasonal time scales; in P12 L30 it is mentioned that these fluctuations may be related to "artifacts that are caused by the sparseness of the observations". It should be investigated as to whether there might be some temporal averaging or aggregation required to reduce the noise.

Authors: Yes, the fluctuations at sub-seasonal time scales are caused by the flux adjustment in each 10-day window, which can be significantly reduced with some temporal averaging, e.g. monthly averaging. We added the smoothed curves to Fig.4. Accordingly, a sentence is added in the revised version, "With monthly averaging, the fluctuations in the derived posterior fluxes could be significantly reduced"

2. Why are the footprints aggregated to 1x1 degrees, given the 10x10 km spatial resolution of WRF resolution, and given the spatial variability of fluxes (including anthropogenic emissions) combined with rather small footprint areas within the proximity of the atmospheric observations? Has any sensitivity analysis been performed to assess the impact of using these rather coarse footprints?

Authors: The footprints are indeed available at higher resolutions, e.g. 10x10 km near the tower sites; however, all major biosphere  $CO_2$  fluxes (SiB3, SiBCASA, CT, CT-Europe) are only available at the resolution of 1x1 degree (~100x100 km), which is especially the case for biosphere OCS fluxes. Since we aim to assimilate both CO2 and OCS with the CTDAS-Lagrange system, we have not made additional efforts to perform sensitivity analysis using higher resolution footprints and higher spatial fluxes than 1 x 1 degrees.

Furthermore, it should be clarified if SiBCASA uses only dominant vegetation types at the 1x1 degree resolution, or whether a tile approach is used to also include other vegetation types within a given grid cell?

Authors: SiBCASA uses one of the 12 dominant biome types at 1x1 degree resolution. But it does include the distinction of C3 and C4 photosynthesis using the C4 coverage map from Still et al. (2003), which means that the grid cells contain a fraction of both C3 and C4 plant types, and the uptake is computed separately from each of the plant types. We added these to the revised version.

3. The shortest length scales used for the prior error covariance are rather large compared to those suggested e.g. by Chevallier et al. (2012) or by Kountouris et al. (2015). Given that the change in the annual net biosphere fluxes with correlation length scale seems to become larger with shorter length scales (Fig. 10), also\_short correlation scales should be investigated. Furthermore, fig. 10 should include the prior flux and the prior and posterior uncertainties (which might change with correlation scale depending on the setup).

Authors: The choice of appropriate correlation length scale depends also on the observation density. For example, CarbonTracker Europe, which includes more observations than those used in this work, uses a correlation length scale of 300 km for North America. Considering the sparseness of the assimilated observations in our study, we think that the tested range of the length scales from 300 km to 1250 km is sufficient. In addition, Alden (2013) found 700 km to be the best length scale to recover true fluxes over North America with a pseudo-data inversion experiment.

We have added the flux uncertainty for both North America and the temperate region (Fig. S1). We need to point out that the formal posterior error estimates in CarbonTracker are always unrealistically large, which is typical for all CarbonTracker applications. In that sense, the spread of multiple inversions with each a different setup has more meaning than the formal uncertainty estimate. This is because for each new 10-day period we need to introduce a fresh prior covariance structure to prevent the uncertainty to converge to zero after several consecutive assimilation cycles, which prevents us to derive meaningful annual mean flux uncertainties.

We have added a few sentences to the revised version.

"The choice of appropriate correlation length scale depends also on the observation density. For example, CarbonTracker Europe, which includes more observations than those used in this work, uses a correlation length scale of 300 km for North America. In addition, Alden (2013) found 700 km to be the best length scale to recover true fluxes over North America with a pseudo-data inversion experiment."

Detailed comments:

P4 L7: Table 1 lists 12 rather than six PFP aircraft sites, also P4 L28 mentions 12 sites

Authors: Thanks for pointing out this inconsistency. Yes, it was a mistake and is corrected in the revised version.

P6 L18: In equation (1) beta\_i, W\_i and S should also have a dependence on observation time t\_r and location X\_r.

Authors: Yes, Wi and S are a function of tr and Xr; however,  $\beta$ i is a constant number within each 10-day period for each of the four sides of the domain, and independent of tr and Xr. We have revised equation (1).

$$C(X_r, t_r) = C_0(X_r, t_r) + \sum_{i=0}^{4} W_i(X_r, t_r) * \beta_i$$

$$+S(X_r, t_r | x, y, t) * \begin{cases} f[\lambda, F_{bio}(x, y, t)] \\ F_{ff}(x, y, t) \\ F_{ocn}(x, y, t) \\ F_{fire}(x, y, t) \end{cases}$$
(1)

We updated the description of the equation in the revised version accordingly.

P6 L23-25: This is unclear. Particles leaving the domain below 3000 m are not considered, does that mean they get a zero boundary condition value for the mole fraction? Even though the influence from surface fluxes is strong, the lateral boundary condition for those should be quite different from zero. What about those particles that did not leave the domain within the 10 days?

Authors: "Particles leaving the domain below 3000 m are not considered" means those were not used to calculate the weight Wi, which is the same for those particles that did not leave the domain within 10 days. When all particles leave the domain below 3000 m, the weights of BC

parameters are zero and the BC will not be adjusted. We added the following sentences in the revised version.

"... which means that the particles that exited the domain below 3000 m or did not leave the domain within 10 days were not used to calculate the weight Wi  $(X_r, t_r)$ . In case all particles left the domain below 3000 m, the weights of BC parameters were zero and the BC was not adjusted"

P7 L24: Why use 3 hourly mean fluxes, as footprints are available at hourly time steps? Was the associated aggregation error quantified? Similar with fossil fuel emissions (P8 L5)

Authors: We based our choice to use 3-hourly fluxes on the availability of ocean, fire, and fossil fuel flux products at this frequency, as also used in the CT2013B runs we use in the sensitivity tests. Hourly fluxes would be preferred, and we think this will be especially important when increasing the resolution of the model beyond the 1x1 degree, 10-day stratification chosen here. This would go hand-in-hand with the use of hourly continuous observations from the tall-tower network, which actually would capture such higher frequency flux signals. Because creating hourly fluxes is not a small task, and we did not currently use the higher resolution model stratification or continuous observations, we did not quantify the associated errors in this work.

P8 L26: "multi-model prior suite of inversion" this is unclear, may be reformulate?

Authors: We changed the sentence to "CT2013B offers a number of flux estimates (ocean, fossil fuels, etc.) from multiple models".

P9 L12: Lateral boundary condition EMP: references only describe LBCs for gases other than CO2. A more detailed description of how EMP CO2 fields were derived is needed. How does the EMP boundary condition differ from the one described in Gerbig et al., 2003?

**Authors:**

Pacific marine boundary layer data from the NOAA Earth System Research Laboratory's Cooperative Air Sampling Network and vertical profile data from aircraft were used to produce a background mole fraction field varying across latitudes, altitudes, and time. This three-dimensional background "curtain" represents mole fractions of CO2 in the remote atmosphere between 10° and 80°N and from 0 to 7500 m above sea level. It was derived using the same curve-fitting algorithms described in Masarie and Tans (1995). Similar background fields have been used in regional inverse-modeling studies of CH4, CO2, and other gases (e.g., Gourdji et al., 2012; Jeong et al., 2013; Miller et al., 2013; Hu et al., 2015).

Similarly, the lateral boundary condition was constructed in Gerbig et al., (2003) based on a series of analytical functions, which were used to fit measurements at selected ground stations and from aircraft data from various campaigns.

**We added these in the revised version.**

P10 L18, Table 2: Cases BX1-BX5 should be included in Table 2 (there only BX2 is included, but the values for the different columns are identical to those for the Base run). Also, the run B2' (included in Table 2) should be described in the text. Similarly, the multiplicative flux adjustment run should be included in Table 2.

**Authors: Done.**

Table 4: the values in brackets need to be explained in the table caption. Also "BC adjustment" should be given as a mean and a range, and the range should be explained in the caption.

**Authors: Done.**

Table 5: the values in brackets need to be explained in the table caption, I assume those are the uncertainties.

**Authors: Done.**

P14 L20: what is meant by "consistent"? Given the uncertainties all flux estimates are statistically indistinguishable.

**Authors: That's right. We change it to "prior biosphere products converge" for better to be understood.**

P14 L23-29: unclear, how the averaging was done. Did the prior fluxes in the CT2013B-avg case have any diurnal variations? The reason for using 10-day averages is also not clear to me. CTDAS-Lagrange uses 10-day backward calculations, but the resulting footprint values change strongly with backward time (time before measurement time).

Authors: Because the prior CT2013B fluxes contain large fluctuations, we have averaged the fluxes within 10-day windows to a single constant value. We are fully aware that this is not realistic, and this should be regarded as a sensitivity test to understanding the difficulties of our CTDAS-Lagrange system to high-frequency fluctuations in the prior fluxes with limited flexibility (prior flux uncertainty). We have added the explanation to the revised version.

P15 L20: OSSEs can be set up in very different ways, allowing also for differences in transport (using different transport models) or structural differences in biospheric fluxes. This should be

**reformulated.**

Authors: The primary aim of our OSSE experiments is to investigate the ability of our system to retrieve surface fluxes given the observational network. In particular, we tested the implementation of the additive flux parameter vs. multiplicative flux parameter, and the ability to recover large biases in lateral boundary conditions and prior fluxes. We have revised the main text to make our intention more clear.

P15 L23, Fig. 8, Fig. 9: Why are the prior uncertainties in the additive method so much larger than in the multiplicative method? For a clear comparison between the two methods, the underlying uncertainties should be better matched. SCALE at which matching is needed?

Authors: It should be noted that the uncertainty for the multiplicative method is changing with the flux magnitude, i.e., the uncertainty assigned to the summer time is much larger than that in winter. For the additive case, uncertainties are kept constant for the entire year. Therefore, the underlying uncertainties cannot be fully matched. We performed a series of tests to increase the uncertainties for the multiplicative method. We included one run with the multiplicative method where we increased the uncertainty by a factor of 2 (added in Figure 9 of the revised manuscript). The fact that no significant change was observed with the increased uncertainty with the multiplicative method indicates that the differences of the optimized fluxes between the two methods were due to the methodology instead of uncertainty settings.

P16 L22: This seems inconsistent with the numbers given in Table 2 and with the description in P11 L22

Authors: The description in P11 L22 and Table 2 are correct, and we changed the text at two places to be consistent with them. Now it reads as "...Q1 (decrease the magnitude of additive uncertainty by 50%) and Q2 (increase the magnitude of additive uncertainty by 50%)"

**References:**

Alden, C. B. (2013). Terrestrial carbon cycle responses to drought and climate stress: new insights using atmospheric observations of CO2 and delta13c. *Dissertations & Theses - Gradworks*.

Chevallier, F., Wang, T., Ciais, P., Maignan, F., Bocquet, M., Altaf Arain, M., Cescatti, A., CHEN, J., Dolman, A. J., Law, B. E., Margolis, H. A., Montag- nani, L. and Moors, E. J.: What eddycovariance measurements tell us about prior land flux errors in CO 2-flux inversion schemes, Global Biogeochem. Cycles, 26(1), doi:10.1029/2010GB003974, 2012.

Gerbig, C., Lin, J.C., Wofsy, S.C., Daube, B.C., Andrews, A.E., Stephens, B.B., Bakwin, P.S. and Grainger, C.A.: Toward constraining regional-scale fluxes of CO2 with atmospheric

observations over a continent: 2. Analysis of COBRA data using a receptor-oriented framework, Journal of Geophysical Research-Atmospheres, 108, 4757, doi:10.1029/2003jd003770, 2003.

Kountouris, P., Gerbig, C., Totsche, K. U., Dolman, A. J., Meesters, A. G. C. A., Broquet, G., Maignan, F., Gioli, B., Montagnani, L. and Helfter, C.: An objective prior error quantification for regional atmospheric inverse applications, Biogeosciences, 12(24), 7403–7421, doi:10.5194/bg-12-7403-2015, 2015.

**Reply to Referee #2**

General comment:

The regional assimilation system presented here is the first semi-operational atmospheric inversion of carbon fluxes at the mesoscale. Compared to previous inversion systems, CTDAS-Lagrange has been significantly improved for high-resolution problems thanks to a state-of-theart atmospheric model and a comprehensive optimization framework including the problem of boundary conditions. The inversion framework is similar to CarbonTracker but has been adapted to the optimization of a pixel-based state vector. The sensitivity experiments performed here shows that the system is capable of reproducing the observed variability in CO2 mixing ratios across North America. The continental fluxes are consistent with global-scale inversions, and the robustness of the optimized surface fluxes to assumptions made in the inversion has been tested carefully. Therefore, we recommend this paper for publication after considering the general and specific comments listed here.

**Authors: We thank the referee for the appreciation of our work for regional inversions. We have improved the manuscript according to the comments below.**

Boundary Conditions: Three products have been used here to describe the CO2 mixing ratios coming from outside the simulation domain. The results show that two of them (i.e. CT2013B and EMP) produce very similar results with nearly identical posterior fluxes. However, the third one (i.e. CTE2014) produces an offset in the late growing season which is not corrected for after inversion. The major concern here is related to the representativity of the three products. The fact that one of them remains significantly different despite the optimization process suggests that the uncertainties in the boundary conditions are not removed by the assimilation of data. Hence, the ensemble has to be representative of the actual uncertainties in boundary conditions to be properly propagated into the flux uncertainties. The authors have not clearly demonstrated that these three products represent the actual errors coming the boundary conditions. This study needs to demonstrate the value of the sensitivity tests and address more carefully the actual error propagation into the posterior fluxes and their uncertainties.

Authors: This is a good point. Our EMP boundary condition is interpolated from smoothed marine boundary layer and aircraft data, which has low mean biases but may not be able to capture large synoptic events. CT2013B and CTE2014 boundaries are optimized CO2 concentration fields but may contain certain biases. We agree that the three boundary condition products are not necessarily representative of the actual uncertainties in boundary conditions; however, they represent a reasonable spread of boundary conditions, including the empirical one that should include no significant mean biases. Furthermore, the fact that the CTE2014 boundary conditions was significantly adjusted in the flux + BC optimization demonstrated the ability of our CTDAS-Lagrange system to correct for the potential biases in the boundary conditions, although the system is not able to correct for all biases.

As explained in the responses to the reviewer#1, a direct error propagation into posterior fluxes is not straightforward, and we would rather include them in an ensemble approach to show the spread of multiple inversion results.

Multiplicative versus additive methods: Because the prior error variances differ significantly between the two experiments, the differences between the two methods depend on the prior error variance more than the actual method to invert the fluxes. The authors need to perform another simulation with similar prior errors to produce convincing evidences that the method used is the fundamental problem.

Authors: We thank the reviewer for this suggestion, which closely follows the remark from reviewer #1. To address this concern we have performed an extra inversion as recommended, using a larger error variance for the multiplicative method to make the summer uncertainty comparable to that in the additive method. We found no significant change when compared with the result from the additive method (details see revised Figure 9), convincing us that indeed the methodology rather than the variances are the main source of difference between these runs.

Correlation length scale in prior errors: In a system with a fairly small degree of freedom such as CTDAS, the spatial attribution of flux corrections may still be sensitive to the definition of the prior errors but the total flux is likely to remain unchanged. The convergence of the system using different length scales can be an artefact due to the degree of freedom but the spatial distribution of flux corrections may vary across the domain. The authors need to include maps of the flux corrections for the different length scales which will show the actual impact of prior error assumptions.

Authors: We added maps of the flux corrections for the different length scales, including those for both annual average and summer average (see revised Figure 10 and Figure S2-4 at the end

of this document). We find that the correlation length scale does not have a significant impact on spatial pattern or on the annual magnitude of optimized fluxes.

Technical comments:

P2-L10: add a reference.

Authors: We have added a citation by Houghton et al (2001).

Houghton, J. T., Ding, Y., Griggs, D. J., Norguer, M., van der Linden, P. J., Dai, X., Maskell, K., and Johnson, C. A.: 2001, 'Climate change 2001: The scientific basis', WGI-Report of the Intergov. Panel on Climate Change, Cambridge University Press, Cambridge, UK, 881 pp.

P2-L11 and L12: Add references to previous studies.

**Authors: Added.**

Atmospheric measurements of trace gas mole fractions provide constraints for the estimates of biosphere surface fluxes from regional to global scales (Schuh et al., 2010; Lauvaux et al., 2012; Peters et al., 2007; Peylin et al., 2013; van der Laan-Luijkx et al., 2017), and complement bottom-up biosphere modeling (Sellers et al., 1996; Schaefer et al., 2008; van der Velde et al., 2014) that typically targets site to ecosystem scales in the Earth system.

P2-L14: This is true for global inversions but domain-limited inversions have opened boundaries. Refine this statement.

Authors: We added "... for global inversions ..." in the revised version.

P2-L22: Feng et al., 2016 is not an inversion study.

Authors: this citation has been removed.

P2-L25: Eulerian models are often a pre-requisite to Lagrangian models, like in CTDAS. You may refer here to the model used in the assimilation framework, typically ensemble-based methods based on Eulerian models and analytical methods with a linearized adjoint model. This statement is unclear for column-based measurements. Refine the statement.

Authors: For regional inversions, Eulerian model results may be needed for the lateral boundary conditions, but are not necessarily needed, e.g. when empirical curtains were used as the lateral boundary conditions. Here we meant to only compare the computation cost due to transport, instead of the assimilation framework. We add the following sentence to the revised version.

"The computation cost of Lagrangian model increases with the increasing number of observations; however, it remains an advantage that offline Lagrangian transport results, i.e. footprints, need to be computed only once, and can be stored for future use."

P2-L32: Peylin et al. (2005) discuss the importance of initial conditions in a global inversion. The problem is different for the lateral flow in a domain-limited inversion which does not decrease over time. Another citation is needed here.

Authors: Thank you for pointing this out. We have replaced the citation with the following.

Andersson, E., Kahnert, M., and Devasthale, A.: Methodology for evaluating lateral boundary conditions in the regional chemical transport model MATCH (v5.5.0) using combined satellite and ground-based observations, Geosci. Model Dev., 8, 3747-3763, https://doi.org/10.5194/gmd-8-3747-2015, 2015

Figure 1: The interpolation for land cover creates artificial zones between land cover types, with halos around temperate Crops / Agriculture for example. The native resolution of the land cover plotted for each grid cell is better suited for land cover maps. Replace contours by actual grid cell colors.

Authors: Thank you for this suggestion. The new figure shows actual grid cell colors.

P4-L21: Indicate the time period over which the flasks have been sampled with the new protocol. How did you treat data with potential biases?

Authors: The new protocol was implemented in September 2013. We did not make any attempt to correct for the potential biases in the data. Masarie et al., 2011 shows that every 1 ppm of bias at LEF in the CarbonTracker inversion causes a linear response rate of 68 Tg C yr-1 for temperate North American net flux estimates. However, if the bias is across the whole network, the impact on the net flux estimates will be much less than that. We added this to the revised version.

P5-L5: Have you considered the possible impact of surface fluxes when sampling at 3km and above? Some of the aircraft sampling locations could be impacted by vertical transport of surface fluxes. Could you use CO to detect high-altitude surface flux influence?

Authors: Yes, the reason why the particles that exited the domain below 3000 m are not considered for calculating the pre-calculated coefficient Wi is to exclude the possible impact of surface fluxes. To a certain degree, CO could be used as a tracer for surface flux influence; however, high CO could also be due to advection from the lateral boundary conditions.

P6-L3: Add a reference to previous studies.

Authors: Added the citation Gerbig et al. (2003).

P6-L7: The description of the WRF model configuration is critical for future users of the footprints. Provide a complete description of the model, as well as the simulation domain (map projection, . . .).

Authors: The WRF model (version 3.3.1 for the 2010 time period of this study) was configured with a Lambert conformal map projection to cover continental North America, with a spatial resolution of 10 km for the inner domain over the continental U.S. (~ 25 - 55 °N; 135 - 65 °W) and 40 km for the outer domain (~ 10 - 80 °N; 170 - 50 °W), and 40 vertical layers, which is similar to the configurations in Nehrkorn et al (2010) and Hu et al (2015). The North American Regional Reanalysis (Mesinger et al. 1996) provided initial and lateral boundary conditions. Model runs were initialized every 24 hours, with the initial 6 hours of each 30-hour forecast discarded to allow for model spinup. We have modified the text in the revised version.

P6-L21: You assume here that the prior BC errors are dominant over 10 days. Is that consistent with model-mismatches? Synoptic systems are more likely to be the main source of errors in the inflow. Justify the 10-day optimization window for boundary conditions.

Authors: Indeed, our current implementation of CTDAS requires the optimization window for fluxes and BC to be the same, i.e. 10 days. We agree that synoptic systems are more likely to be the main source of errors in the inflow and it is actually a nice suggestion by the reviewer to perhaps optimize BCs in steps of, say, 5-days. However, that also would require us to change the structure of the BC parameters to allow for synoptic spatial patterns that do not stratify into the simple 4-sides-to-a-box approach demonstrated here. Considering the influence of the BCs it would be good to investigate this in more detail, but considering the amount of technical innovations we report on here already, we would prefer to leave it as future development of the system.

P6-L24: You assume here that the errors in the BC's are identical below and above 3000m. This assumption seems very unlikely as the modeled values near the surface will differ significantly from the modeled mixing ratios in the Free Troposphere. An evaluation of the model-data mismatches compared to the altitude would help infer the actual vertical structures in BC errors.

Authors: We would like to point out that the setup of using four parameters for the BC optimization within a 10-day window is a first attempt to simultaneously optimize surface fluxes and BCs. Assigning different errors in the BC for below and above 3000 m could be an intermediate step to improve the BC optimization, compared with e.g. a full grid optimization of

the boundary conditions. Similar to the response to the previous question, we would leave it as future development work.

**P8-L2: Which observations are assimilated here? Daily afternoon averages?**

Authors: The observations include both tall tower and aircraft observations, which have been described in Sect. 2.1. Air samples were collected daily or on alternate days during midafternoon at the tall tower sites, and biweekly or monthly at the selected aircraft sites. We use daytime data from the tall towers that are collected between 10:00 and 18:00 local time to constrain surface fluxes. Aircraft observations made at altitudes higher than 3000 meters above ground at all hours are used to constrain boundary conditions.

P11-L18: Justify the removal of outliers here. The factor of 3 applied to the MDM is arbitrary. Any physical reason behind this?

Authors: We reject the outliers that exceed 3 times the prescribed MDM as an indicator of the moments either our modeling framework is failing or when a bad observation is entering the filter (and thus rejected). That means according to the 3-sigma criterion about 2% is removed. This criterion is used for most CarbonTracker applications in the past, and we actually partly chose our model-data mismatches to remain close to this percentage at each site. It has been mentioned in the main text "It is shown in Table 1 that the rejection rates for most tower sites are around 2-3%, except for WBI (7.6%) and WGC (18.0%)."

P12-L25: The reduction in uncertainty is a direct consequence of the optimization process but does not mean that the actual errors are reduced. Similar to the posterior CO2 mixing ratio mismatches, the optimization was designed to reduce it, except that fluxes have no guarantee that the reduction is real. Clarify in the paragraph.

Authors: The reviewer is correct to say that formal posterior variance is reduced by design of an inversion, and does not necessarily mean that a better answer is obtained. That is why we prefer (as also indicated in our response to reviewer #1) to use the spread of multiple inversions to assess the errors that remain after optimizations. To clarify it, we have added the following sentence.

"It should be noted that it does not mean that the actual errors in these fluxes are really reduced, as this can only be assessed using independent observations of these fluxes."

Figure 4: The aggregation of large ecoregions over the entire continent makes the interpretation of the results very difficult. The ecoregions are too wide to make sense of the seasonal cycles that correspond to multiple regions and climates. The separation of the results into smaller regions is needed here. For example, southern and northern regions for crops

would provide a better sense of what the seasonal should look like. This figure needs to represent the ecosystems defined in climate zones.

Authors: Indeed, the same ecoregions may correspond to multiple regions and climates. As suggested, we have separated the southern and northern regions for crops and forests (divided by 40°N), and updated Figure 4. The seasonal cycles are mainly caused by those in the northern regions, especially for crops.

We have added the following sentence to the revised version.

"Since the same ecoregions may correspond to multiple regions and climates, we have separated the southern and norther regions for crops and forests (divided by 40°N). The seasonal cycles are mainly caused by those in the northern regions, especially for crops"

P13-L26: - Except for CTE2014, the adjustment to the boundary conditions converges to identical posterior flux values for CT2013B and EMP. These two products are fundamentally different, but the correction to the BC's are significantly different, which suggests that the inversion is able to correct and hence reduce the initial differences between the two. However, CTE2014 produces a different result, which would suggest that the BC errors are too small, or that the lack of data limits the ability of CTDAS to reduce the BC mismatches. Provide more elements here to understand what caused the final disagreement between CTE2014 and the two other BC's. Even for the CTE2014 case, the limited impact on the optimized surface fluxes is also surprising. Less than 0.25 PgC/yr is small compared to the large posterior uncertainties (about 1.7 PgC/yr). Is your ensemble truly representative of the BC errors? The BC's need to be illustrated here with the differences across the different products before inversion. Is the simplification into four boundaries too limiting in the optimization?

Authors: This is a good point. We have performed a new set of sensitivity runs with increased BC parameter uncertainties (2 ppp, 3 ppm and 4 ppm) and the CTE2014 boundary condition. The resulting total annual fluxes are  $-1.260 \pm 1.757$ ,  $-1.237 \pm 1.766$  and  $-1.216 \pm 1.766$  PgC/yr, respectively, which indicates that increasing the BC parameter uncertainties results in slightly smaller annual fluxes, but not significant enough to explain the deviation from the posterior fluxes for CT2013B and EMP. Besides these, we do think the simplification into four boundaries and within a 10-day window also limit the optimization. As we have answered to a similar question from reviewer 1, we would prefer to leave the further improvement of the boundary condition optimization as future development of the system. See also our earlier response to the representativeness of the BC uncertainties.

Section 3.5: The flux adjustment compares two different prior errors with significantly different values in the shoulder season. Therefore, the methods disagree mainly when the prior errors are too small to let the inversion adjusts the fluxes. The comparison is presented based on the nature of the correction, but it turns out to illustrate the importance of larger errors when the

NEE value is small, and hence the scaled prior variances are small as well. Here, you illustrated the dependence to prior error variances instead of a method-dependence.

Authors: The major difference between the two methods "multiplicative versus additive" is that the error variances are prescribed differently. See also our responses to the comment from Reviewer #1 on the discussion of the uncertainty of the two methods.

Section 3.6: This exercise requires a map of the posterior corrections. Assuming that the number of towers is sufficient to constrain most of North America, the total carbon budget will remain fairly similar. It also confirms that your degree of freedom is small, as described earlier in the paper. In this case, changing the correlation length scale will not lead to any significant changes because the inversion is over-constrained. Only biases will alter the results, such as systematic offsets in the BC. A figure illustrating the spatial distribution of flux corrections is needed here.

Authors: See our response to the previous question "Correlation length scale in prior errors"

We thank the two referees again for thorough and helpful reviews, which have undoubtedly improved our manuscript.

**Supplemental figures**

Figure S1. Sensitivity of the optimized annual net biosphere fluxes (PgC/yr) as a function of the chosen covariance length scale (km): the left panel shows fluxes and uncertainty of continental North America, and the right panel shows fluxes and uncertainty of temperate North America. The optimized fluxes tend to converge to -1.1 PgC/yr when the length scale is larger than 750 km. (better noticeable in Figure 10 in the revised manuscript)